# LoopTool: Closing the Data–Training Loop for Robust LLM Tool Calls

## Abstract

Augmenting Large Language Models (LLMs) with external tools enables them to execute complex, multi-step tasks. However, tool learning is hampered by the static synthetic data pipelines where data generation and model training are executed as two separate, non-interactive processes. This approach fails to adaptively focus on a model's specific weaknesses and allows noisy labels to persist, degrading training efficiency. We introduce **LoopTool**, a fully automated, model-aware data evolution framework that closes this loop by tightly integrating data synthesis and model training. LoopTool iteratively refines both the data and the model through three synergistic modules: (1) *Greedy Capability Probing (GCP)* diagnoses the model's mastered and failed capabilities; (2) *Judgement-Guided Label Verification (JGLV)* uses an open-source judge model to find and correct annotation errors, progressively purifying the dataset; and (3) *Error-Driven Data Expansion (EDDE)* generates new, challenging samples based on identified failures. This closed-loop process operates within a cost-effective, open-source ecosystem, eliminating dependence on expensive closed-source APIs. Experiments show that our 8B model trained with LoopTool significantly surpasses its 32B data generator and achieves new state-of-the-art results on the BFCL-v3 and ACEBench benchmarks for its scale. Our work demonstrates that closed-loop, self-refining data pipelines can dramatically enhance the tool-use capabilities of LLMs.[1]

## 1 Introduction

Large Language Models (LLMs) augmented with external tools have become a powerful paradigm for solving complex tasks beyond pure text generation (Qu et al., 2025; Schick et al., 2023; Qin et al., 2023). By invoking APIs, querying databases, and interacting with computational engines, such agents can tackle diverse real-world scenarios (Chen et al., 2025b; Xie et al., 2024; Pan et al., 2025) with high efficiency and adaptability. The development of robust tool-use capabilities, however, hinges on access to accurate, large-scale, and well-aligned training data that matches the model's current competencies (Liu et al., 2025).

A widely adopted approach in this domain involves constructing large-scale tool-calling datasets through automated synthesis pipelines (Qin et al., 2023; Liu et al., 2024; Tang et al., 2023; Liu et al., 2025; Prabhakar et al., 2025), followed by supervised fine-tuning (SFT) or reinforcement learning (Wang et al., 2025; Shao et al., 2024). Despite notable advances, **they almost invariably adopt a static design, wherein data generation and model training are executed as two separate, non-interactive processes**. In these settings, the training data is generated *a priori* without awareness of the evolving state of the model, causing wasted capacity on trivial cases already mastered while leaving harder, underrepresented cases unresolved. Furthermore, the model plays no role in guiding or influencing data generation. This inherent disconnect leads to a persistent mismatch between the model's learning needs and the fixed nature of the available training data, thereby constraining both the efficiency and effectiveness of post-training.

Another major challenge in tool-use data generation lies in the trade-off between cost-efficiency and data quality. Many pipelines depend on large closed-source models (OpenAI, 2023) for data generation and evaluation. While these models are capable of producing high-fidelity tool-calling

---

[1]The code is accessible in this anonymous repository https://anonymous.4open.science/r/LoopTool.

sequences, their use incurs high API costs and low generation efficiency, making frequent large-scale data synthesis impractical. Replacing them with more accessible open-source models often introduces noisy annotations, including incorrect arguments, incomplete function calls, or outputs misaligned with task requirements. Such errors inject misleading learning signals and can undermine model generalization (Liu et al., 2025).

To address the limitations of static, costly, and error-prone tool-use data pipelines, we propose Loop-Tool—an automatic, model-aware data evolution framework that couples data synthesis and training in a closed loop. LoopTool begins with an Automated Tool-Augmented Data Construction stage, where tool specifications are synthesized and combined with multi-agent dialogue generation to produce a diverse seed corpus of realistic tool-oriented conversations. This corpus undergoes an initial GRPO-based (Shao et al., 2024; DeepSeek-AI et al., 2025) post-training round.

Each iteration then integrates three synergistic modules. First, **Greedy Capability Probing (GCP)** queries the fine-tuned model on the seed corpus using greedy decoding, revealing mastered, borderline, and failure cases. The predicted tool calls are used for automated error analysis, allowing the pipeline to target challenging, underperforming cases. Second, **Judgement-Guided Label Verification (JGLV)** employs a high-capacity open-source judge model, Qwen3-32B (Yang et al., 2025), to compare each prediction against its reference label—identifying genuine model errors as well as cases where the model output surpasses the original annotation. Such "model-better-than-label" examples replace noisy labels, enabling systematic self-refinement and progressively purifying the supervision signal. Third, **Error-Driven Data Expansion (EDDE)** transforms verified failure cases into new, structurally similar but contextually diverse challenging samples. Augmented samples preserve the core functional challenge while introducing varied conditions, ensuring scenario diversity. Across iterations, LoopTool incorporates corrected annotations, diversified hard samples, and refined seeds into subsequent training rounds, creating a dynamic curriculum attuned to the model's evolving strengths and weaknesses. This process focuses learning on non-trivial, high-value opportunities while progressively mitigating noisy-label effects.

To balance quality and cost, LoopTool unifies the roles of data generator and evaluation judge within a single, open-source model, Qwen3-32B, eliminating reliance on expensive closed-source APIs while maintaining high data quality. Strikingly, despite being trained entirely on data generated and evaluated by Qwen3-32B, the final 8B-scale LoopTool model surpasses the 32B generator in tool-use performance, highlighting the amplifying effect of iterative, model-aware data refinement.

In summary, our main contributions are:

- We present **LoopTool**, the first fully automated, model-aware iterative framework that tightly couples reinforcement learning post-training with targeted data synthesis. By continuously diagnosing model weaknesses and generating capability-targeted training data, LoopTool enables dynamic **co-adaptation** of the model and the dataset.

- We propose a closed-loop data refinement and augmentation strategy that purifies labels through comparative judgment (**JGLV**) and transforms verified failures into diverse, high-value training samples (**EDDE**), enhancing tool-use learning without reliance on closed-source models.

- Leveraging fully open-source, self-contained data generation and refinement, our 8B model trained by LoopTool surpasses its 32B generator and achieves state-of-the-art performance on BFCL-v3 (Patil et al., 2025) and ACEBench (Chen et al., 2025a) among models of similar scale.

## 2 RELATED WORK

**Tool-Augmented Large Language Models.** Integrating large language models (LLMs) with external tools has proven effective in overcoming their inherent limitations (Qu et al., 2025).Such integration enables API invocation (Shen et al., 2023; Qin et al., 2023), interaction with knowledge bases (Lazaridou et al., 2022; Chen et al., 2025b), code execution (Wang et al., 2024), and multimodal processing (Hu et al., 2024; Ma et al., 2024). Early efforts mainly relied on supervised fine-tuning (SFT) with human-labeled tool-use data, focusing on accurate tool selection and argument generation (Schick et al., 2023; Qin et al., 2023; Liu et al., 2024). Recent advances explore autonomous tool creation and dynamic invocation, enabling adaptation to unseen APIs without predefined schemas. Benchmarks such as tau-bench (Yao et al., 2024; Barres et al., 2025), BFCL (Patil

et al., 2025), and ACEBench (Chen et al., 2025a) provide standardized evaluations across tool selection, argument generation, multi-step reasoning, and multi-turn tool calling.

**Synthetic Data Generation for Tool Use.** The scarcity and cost of high-quality tool-use datasets have driven research into automated synthesis pipelines (Qin et al., 2023; Liu et al., 2025; 2024; Prabhakar et al., 2025). Methods include multi-agent simulation (Alvarez et al., 2024; Tang et al., 2024), modular task composition (Chen et al., 2025c), and graph-based query–function synthesis (Arcadinho et al., 2024; Yin et al., 2025). Our work builds on this line but differs by introducing a fully automated, model-aware, iterative paradigm in which synthesis is guided by post-training diagnostics and refined via systematic error correction.

**Reinforcement Learning for Tool-Use Optimization.** Reinforcement learning (RL) increasingly enhances LLM reasoning and decision-making (Ouyang et al., 2022; Rafailov et al., 2024; Meng et al., 2024; Shao et al., 2024). In tool-use settings, GRPO has shown strong performance (Qian et al., 2025; Zhang et al., 2025). We embed RL into an interleaved train–generate loop, enabling the model to iteratively improve through exposure to prior failures and progressively refined supervision.

# 3 AUTOMATED TOOL-AUGMENTED DIALOGUE CONSTRUCTION

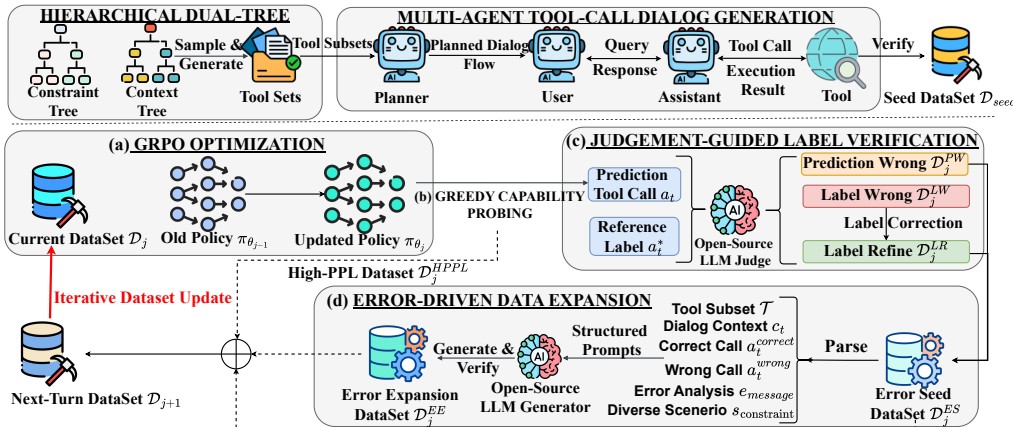

Figure 1: The overall closed-loop automatic pipeline of **LoopTool**, which couples (a) GRPO optimization, (b) Greedy Capability Probing, (c) Judgement-Guided Label Verification, and (d) Error-Driven Data Expansion for iterative tool-use enhancement.

Before initiating our iterative model-aware data evolution process, we require a diverse and high-quality *seed dataset* $\mathcal{D}_{\text{seed}}$ to support the first round of post-training. To this end, we introduce an **Automated Tool-augmented Data Construction** that synthesizes realistic function-calling interactions by combining curated APIs with simulated multi-agent conversations. While this stage is *not* the core innovation of our work, it establishes the essential foundation for the following iterations.

## 3.1 HIERARCHICAL DUAL-TREE GUIDED API SYNTHESIS

Our tool set comprises both real-world APIs collected from public resources (Liu et al., 2025; 2024) and synthetically generated APIs produced via a *Hierarchical Dual-Tree* method. For each application domain, we define two complementary hierarchical structures: (i) **Context Tree** encodes the topical scope and functional granularity of the domain, from coarse categories at the root to fine-grained specializations at the leaves; (ii) **Constraint Tree** specifies structural and functional constraints for valid APIs, such as naming conventions, parameter types and counts, and output formats. To synthesize an API, we independently sample a leaf path from each tree and merge the results into a structured prompt for the LLM, ensuring that both functional intent and structural requirements are satisfied. Rule-based validation is subsequently applied to ensure conformity and semantic coherence. Concrete examples of Context and Constraint Trees are provided in Appendix G.

## 3.2 MULTI-AGENT TOOL-USE DIALOG GENERATION

The dialog generation stage incorporates two components: the Multi-Agent Dialogue Simulation and Correctness Verification for quality control.

**Multi-Agent Dialogue Simulation.** We populate the seed dataset by simulating tool-usage dialogues with four distinct roles: *Planner Agent* designs coherent conversation flows based on a sampled subset of tools and a target number of dialog turns. This planning phase ensures realistic task decomposition and natural progression toward tool use. *User Agent* interacts with the assistant according to the Planner's high-level outline, generating new requests, clarifying requirements, or providing additional information such as missing parameters. *Assistant Agent* selects appropriate APIs from the assigned subset, extracts candidate parameters based on the dialog context, executes tool calls, or synthesizes responses for the user. *Tool Agent* processes the tool calls according to the given API definitions and produces simulated execution results. For certain domains, we integrate real executable backends to return authentic responses through actual code execution. The dialog proceeds turn-by-turn until the predefined conversation length is reached.

**Rule-based and LLM-based Verification.** All generated dialogues undergo a two-tier verification process. Rule-based verification checks API call syntax, parameter coverage, type matching, and adherence to schema definitions. LLM-based evaluation leverages an open-source judge model (Qwen3-32B) to holistically evaluate every tool call step for contextual appropriateness, logical consistency, and alignment with the user's intent. Only dialogues satisfying both stages are admitted into the initial seed dataset.

## 4 ITERATIVE MODEL TRAINING AND DATA AUGMENT

To overcome the limitations of static data generation and support dynamically adaptive model training, we develop an automated iterative framework for tool-augmented LLM learning as shown in Figure 1. LoopTool integrates the GRPO Optimization, Greedy Capability Probing, Judgement-Guided Label Verification, and Error-Driven Data Expansion into a unified closed loop. This iterative cycle enables the model to assess its own capabilities continuously, target its weaknesses, and refine the quality of supervision data.

### 4.1 GRPO TRAINING FOR TOOL CALLING

**Data Format.** We construct an **initial seed tool-calling dialogue dataset** $\mathcal{D}_{\text{seed}}$ through the Automated Tool-Augmented Data Construction in Section 3. Each multi-turn dialog sample is transformed into multiple GRPO training samples, which consist of the tuple: $(\mathcal{T}, c_t, a_t^*)$, where $t$ denotes the current turn in the dialogue, as a single conversation may contain multiple sequential tool calls. $\mathcal{T}$ denotes the set of available tools at the current step, $c_t$ represents the historical dialogue context, which can be either a single-turn user query or a multi-turn conversation. $a_t^*$ is the tool call step from the conversation corresponding to the last user query. The model's output $O_t$ include two structured components: a reasoning trace wrapped within `<think>...</think>` and the predicted tool invocation $a_t$ inside `<tool_call>...</tool_call>`. A detailed specification of both the single-turn and multi-turn training formats is provided in Appendix H.

**Binary Reward Definition.** To quantify the quality of model-generated tool calls, we adopt a *Binary Reward* scheme, which serves as a simple yet effective rule-based reward function. For a given context $c_t$ and the model output $a_t$, the reward is defined as:

$$r(\mathcal{T}, c_t, a_t^*, a_t) = \begin{cases} 1, & \text{ToolMatch}(a_t, a_t^*) \\ 0, & \text{otherwise} \end{cases} \tag{1}$$

**GRPO Optimization.** Given the tool sets $\mathcal{T}$ and historical dialogue $c_t$, the policy $\pi_\theta$ sample a group of candidate response $\{O_t^1, O_t^2, \ldots, O_t^G\}$ from the old policy $\pi_{\theta_{\text{old}}}$ and their corresponding rewards are $\{r_t^1, r_t^2, \ldots, r_t^G\}$. We optimizes the $\pi_\theta$ through maximizing the following objective:

$$\mathcal{J}_{\text{GRPO}}(\theta) = \mathbb{E}_{(\mathcal{T}, c_t) \sim \mathcal{D}, \{O_t^i\}_{i=1}^G \sim \pi_{\theta_{\text{old}}}} \frac{1}{G} \sum_{i=1}^G \left[ \min \left( \rho_t^i A_t^i, \, \text{clip}(\rho_t^i, 1 - \epsilon, 1 + \epsilon) A_t^i \right) - \beta \, \text{KL}(\pi_\theta \, \| \, \pi_{\text{old}}) \right],$$

$$\text{where } \rho_t^i = \frac{\pi_\theta(O_t^i \mid c_t, \mathcal{T})}{\pi_{\theta_{\text{old}}}(O_t^i \mid c_t, \mathcal{T})}, \quad A_t^i = \frac{r_t^i - \text{mean}(\{r_t^1, r_t^2, \ldots, r_t^G\})}{\text{std}(\{r_t^1, r_t^2, \ldots, r_t^G\})} \tag{2}$$

$\epsilon$ is the PPO clipping parameter, and $\beta$ controls the strength of the KL penalty.

## 4.2 GREEDY CAPABILITY PROBING

GRPO-based post-training often assigns near-zero advantage values to both trivially solvable and prohibitively hard samples, resulting in negligible parameter updates despite non-trivial computational costs (Yu et al., 2025). To mitigate this inefficiency, we introduce **Greedy Capability Probing** (GCP)—an offline diagnostic stage to identify samples of substantive learning value.

Given the training set $\mathcal{D}_j$ in the $j$-th iteration, we perform deterministic greedy decoding with the current policy $\pi_{\theta_j}$ on every instance. For each tool-call sample $(\mathcal{T}, c_t, a_t^*)$, the model generates a prediction $a_t \in O_t$ via greedy search. If $a_t = a_t^*$, the sample is provisionally considered *mastered* under the assumption that its label is correct. Otherwise, the quadruple $(\mathcal{T}, c_t, a_t^*; a_t)$ is passed to **Judgement-Guided Label Verification** (JGLV) for correctness assessment. To further quantify sample difficulty, we compute sample-level perplexity(PPL) as:

$$\text{PPL}_{(\mathcal{T}, c_t)} = \exp \left( -\frac{1}{L} \sum_{i=1}^L \log p_\theta(o_i \mid \mathcal{T}, c_t, o_{1:i-1}) \right) \tag{3}$$

where $L$ is the output length and $o_i$ denotes the $i$-th token in the output sequence. High perplexity indicates low model confidence and suggests that the sample resides near the decision boundary, making it more valuable for continued training. In subsequent iterations, GCP selectively retains a subset of these high-PPL cases $\mathcal{D}_j^{\text{HPPL}}$ into the next-turn iteration.

## 4.3 JUDGEMENT-GUIDED LABEL VERIFICATION

To mitigate the impact of noisy synthetic annotations and integrate *automatic label refinement* directly into the iterative loop, we introduce **Judgement-Guided Label Verification (JGLV)**—a structured evaluation stage that distinguishes genuine model failures from annotation errors.

In each iteration $j$, for every mismatched case $(\mathcal{T}, c_t, a_t^*; a_t)$ identified by **Greedy Capability Probing**, we organize the tool specifications $\mathcal{T}$, dialogue context $c_t$, reference label $a_t^*$ and model prediction $a_t$ into an open-source LLM—in our implementation, *Qwen3-32B* (Yang et al., 2025)-which outputs a categorical decision: $y_{\text{judge}} \in \{\text{PRED\_WRONG}, \text{LABEL\_WRONG}, \text{BOTH\_CORRECT}, \text{BOTH\_WRONG}\}$ and formatted error analysis $e_{\text{message}}$. Based on the judgment results, we define two key subsets of the evolving dataset: the Prediction Wrong set and the Label Wrong Set.

$$\mathcal{D}_j^{PW} = \{(\mathcal{T}, c_t, a_t^*; a_t) \mid y_{\text{judge}} = \text{PRED\_WRONG}\}$$
$$\mathcal{D}_j^{LW} = \{(\mathcal{T}, c_t, a_t^*, a_t) \mid y_{\text{judge}} = \text{LABEL\_WRONG}\} \tag{4}$$

$\mathcal{D}_j^{PW}$ are retained for retraining in the next iteration. We replace the $a_t^*$ in $\mathcal{D}_j^{LW}$ with $a_t$ to transform the dataset into $\mathcal{D}_j^{LR}$(Refer to Appendix I for judgement prompt and detailed samples). For samples classified as BOTH\_CORRECT, we retain only those with high-PPL into $\mathcal{D}_j^{\text{HPPL}}$. Samples identified as BOTH\_WRONG are directly discarded to avoid propagating noisy supervision.

Compared with approaches that rely on a large language model to directly regenerate or correct labels, JGLV reframes annotation refinement as a *comparative judgment task*, where the judge model only determines which of two existing candidates better satisfies the task specification instead of producing a new output from scratch. Moreover, by incorporating outputs from the evolving current policy into the judgment process, **JGLV leverages the model's progressively improving tool-calling competence to assist data refinement.** As training advances, the policy increasingly produces

valid and high-quality tool invocations, enabling the replacement of incorrect labels with superior model outputs. This synergy transforms label verification into a self-reinforcing mechanism that continuously generates cleaner and more representative training data.

### 4.4 Error-Driven Data Expansion

While GCP and JGLV effectively identify mismatched cases and correct noisy labels, reusing these instances without modification often yields marginal benefit (see Section 5.4), especially when failures arise from systematic weaknesses rather than incidental noise. To directly broaden the model's coverage of challenging tool-use scenarios, we propose Error-Driven Data Expansion (EDDE)—an augmentation strategy that transforms verified failure cases into structurally similar "hard" samples.

In iteration $j$, EDDE operates on the union of the $\mathcal{D}_j^{MR}$ and $\mathcal{D}_j^{LR}$ identified by JGLV: $\mathcal{D}_j^{ES} = \mathcal{D}_j^{MR} \cup \mathcal{D}_j^{LR}$. For each error seed $(\mathcal{T}, c_t, a_t^*; a_t) \in \mathcal{D}_j^{ES}$, EDDE parses the following structured components: tool subset $\mathcal{T}$, dialog context $c_t$, correct call $a_t^{\text{correct}}$, wrong call $a_t^{\text{wrong}}$, and error analysis $e_{\text{message}}$. The generator is instructed to produce $k$ new tool-calling samples that mirror the structural complexity of the error seed (e.g., similar argument, multi-step dependencies). To avoid excessive similarity among the augmented samples derived from the same error seed, we additionally introduce scenario diversification constraints $s_{\text{constraint}}$. Specifically, each generation prompt is enriched with varied situational contexts—such as alternative user goals, different domain-specific constraints, or modified environmental conditions—while preserving the core challenge (Refer to Appendix J for error generation prompt and new generated samples). All EDDE-generated samples are subjected to the same two-tier validation pipeline outlined in Section 3.2—including rule-based and LLM-based evaluation. Samples passing both filters are collected into: $\mathcal{D}_j^{EE} = \text{Verify}\big(\text{Generate}(\mathcal{D}_j^{ES})\big)$.

**Integration into the Iterative Loop.** At the end of iteration $j$, the training dataset for the next round is constructed by merging multiple sources identified during the current iteration:

$$\mathcal{D}_{j+1} = \mathcal{D}_j^{ES} \cup \mathcal{D}_j^{EE} \cup \mathcal{D}_j^{\text{HPPL}} \cup \mathcal{D}_j^{\text{Seed-new}} \tag{5}$$

where $\mathcal{D}_j^{\text{Seed-new}}$ is a small untrained subset from the initial seed dataset $\mathcal{D}_{\text{seed}}$. This merged dataset $\mathcal{D}_{j+1}$ is then used in the subsequent GRPO training round, with the policy $\pi_{\theta_j}$ serving as the initialization. The full iteration pipeline is summarized in the Algorithm 1.

## 5 Experiments

Table 1: Comprehensive evaluation of the BFCL-v3 (last updated on 2025-06-14). FC denotes that the model is tailored for functional calling. The best results in each category are highlighted in bold, while the second-best are underlined.

| Rank | Overall Acc | Model | Single-Turn | | Multi-Turn | Hallucination | |
| | | | Non-Live AST Acc | Live Acc | Overall Acc | Relevance | Irrelevance |
|---|---|---|---|---|---|---|---|
| 1 | 78.45 | xLAM-2-70b-fc-r (FC) | 88.44 | 72.95 | **75.00** | 66.67 | 78.91 |
| 2 | 76.43 | xLAM-2-32b-fc-r (FC) | 89.27 | 74.23 | 67.12 | 88.89 | 76.74 |
| **3** | **74.93** | **LoopTool-8B (Ours)** | **89.52** | **84.72** | 50.88 | 61.11 | 87.67 |
| 4 | 73.57 | watt-tool-70B (FC) | 84.06 | 77.74 | 58.87 | **94.44** | 76.32 |
| 5 | 72.04 | xLAM-2-8b-fc-r (FC) | 84.40 | 66.90 | 69.12 | 77.78 | 64.34 |
| 6 | 71.71 | GPT-4o-2024-11-20 (FC) | 86.81 | 78.85 | 50.00 | 83.33 | 81.31 |
| 7 | 70.42 | GPT-4o-2024-11-20 (Prompt) | 87.67 | 79.88 | 43.00 | 72.22 | 85.36 |
| 8 | 70.32 | GPT-4.5-Preview-2025-02-27 (FC) | 86.12 | 79.34 | 45.38 | 66.67 | 83.64 |
| 9 | 69.25 | Qwen3-32B (FC) | 88.90 | 77.83 | 43.12 | 72.22 | 75.79 |
| 10 | 68.89 | GPT-4.1-2025-04-14 (FC) | 85.42 | 79.92 | 40.50 | 77.78 | 85.95 |
| 11 | 68.73 | ToolACE-2-8B (FC) | 87.58 | 80.05 | 37.00 | 72.22 | **90.11** |
| | | ... (Ranks 12–18 omitted for brevity) | | | | | |
| 19 | 66.34 | Qwen3-8B (FC) | 88.81 | 78.54 | 33.00 | 77.78 | 79.08 |
| 20 | 65.19 | Qwen3-8B (FC, **self-host**) | 87.06 | 78.50 | 31.25 | 77.78 | 78.74 |

### 5.1 Experiment Setup

**Benchmarks.** We evaluate LoopTool by training LLMs with our data generation pipeline, using the open-source Qwen3-8B (Yang et al., 2025) as the backbone under pure RL fine-tuning[2]. Experiments are conducted on two representative benchmarks: BFCL-v3 (Patil et al., 2025) and ACEBench (Chen et al., 2025a), which provide comprehensive, executable function-call tasks for

---

[2]We also investigate the performance of the LoopTool framework on another model architecture, Llama-3.1-8B-Instruct; detailed results are provided in the Appendix E.

Table 2: Comprehensive evaluation of ACEBench for English Data (last updated on 2025-07-21). LoopTool-8B (Ours) achieves the best result in the 8B scale.

| Model | Normal | | | | | | Special | Agent | Overall |
|---|---|---|---|---|---|---|---|---|---|
| | Atom | Single-Turn | Multi-Turn | Similar API | Perference | Summary | | | |
| *Closed-Source Large Language Models* | | | | | | | | | |
| GPT-4o | 90.0 | 78.0 | 68.0 | 80.0 | 78.0 | 82.5 | 92.7 | 56.0 | **81.1** |
| Gemini-2.5-Pro-05-06 | 83.7 | 73.5 | 61.0 | 72.0 | 58.0 | 75.1 | 90.7 | 52.5 | **75.8** |
| Qwen-Max | 88.0 | 75.0 | 61.0 | 74.0 | 82.0 | 79.7 | 74.0 | 60.0 | **75.1** |
| GPT-4o-Mini | 84.3 | 73.5 | 59.0 | 74.0 | 72.0 | 76.4 | 76.7 | 27.5 | **68.9** |
| Gemini-1.5-Pro | 82.3 | 73.0 | 61.0 | 74.0 | 72.0 | 75.7 | 77.3 | 26.0 | **68.5** |
| Claude-3-5-Sonnet | 66.7 | 64.0 | 46.0 | 58.0 | 68.0 | 62.2 | 72.7 | 44.0 | **62.2** |
| Doubao-Pro-32k | 75.3 | 58.0 | 52.0 | 70.0 | 54.0 | 66.3 | 50.7 | 26.5 | **56.0** |
| *Open-Source Large Language Models* | | | | | | | | | |
| Kimi-k2-0711 | 87.0 | 78.5 | 62.0 | 70.0 | 74.0 | 78.9 | 81.3 | 65.0 | **77.4** |
| Qwen2.5-Coder-32B-Instruct | 86.0 | 73.5 | 59.0 | 76.0 | 72.0 | 77.4 | 80.0 | 50.0 | **73.9** |
| **LoopTool-8B (Ours)** | **86.0** | **76.0** | **58.0** | **74.0** | **78.0** | **78.0** | **80.7** | **43.3** | **73.4** |
| ToolACE-2.5-Llama-3.1-8B | 87.7 | 75.5 | 62.0 | 74.0 | 66.0 | 78.3 | 76.0 | 35.9 | **71.1** |
| DeepSeek-V3 | 88.0 | 77.5 | 63.0 | 76.0 | 78.0 | 80.3 | 72.7 | 34.0 | **71.1** |
| Qwen2.5-72B-Instruct | 81.3 | 74.5 | 64.0 | 76.0 | 80.0 | 76.8 | 74.0 | 37.5 | **70.0** |
| Qwen3-8B | 80.3 | 68.5 | 52.0 | 70.0 | 58.0 | 70.9 | 78.0 | 34.2 | **67.1** |
| Llama-3.1-70B-Instruct | 83.7 | 71.5 | 61.0 | 74.0 | 66.0 | 75.6 | 29.3 | 41.0 | **57.9** |
| Qwen2.5-7B-Instruct | 70.3 | 57.0 | 49.0 | 62.0 | 58.0 | 62.8 | 49.3 | 15.0 | **51.8** |
| Qwen2.5-Coder-7B-Instruct | 73.3 | 63.5 | 52.0 | 70.0 | 58.0 | 66.6 | 25.3 | 18.5 | **48.1** |

assessing function invocation capability. We also perform ablation studies to examine the contribution of individual modules. Benchmark details and evaluation metrics are provided in Appendix B.1.

**Implementation Details.** GRPO training is implemented with the open-source RL library Verl (Sheng et al., 2025), using a batch size of 128 and a learning rate of $1 \times 10^{-6}$. Each iteration trains for two epochs, resetting optimizer parameters while initializing from the previous checkpoint. To promote exploration, the actor rollout temperature is fixed at 1.0, with both entropy coefficient and KL weight set to 0. We apply the Clip-Higher (Yu et al., 2025) strategy, increasing $\mathcal{E}_{high}$ from 0.2 to 0.28 to encourage generation of high-entropy, low-probability tokens. In EDDE, $k$ is set to 4. Full hyperparameters are listed in Appendix B.2.

## 5.2 OVERALL PERFORMANCE ANALYSIS

**Result on BFCL and ACEBench.** We compare LoopTool-8B model with various representation models in BFCL (Patil et al., 2025) and ACEBench (Chen et al., 2025a). We adopt the official evaluation script and report the average accuracy across categories. The results are summarized in Table 1 and Table 2, respectively. On both BFCL-v3 and ACEBench leaderboards, LoopTool-8B achieves SOTA performance among all 8B-scale open-source models and exceeds several larger counterparts. In BFCL-v3 (Table 1), our model attains an overall accuracy of **74.93%**, ranking third across all models and surpassing the original Qwen3-8B by **+8.59 points**, with the highest Single-Turn and Live execution accuracy. Remarkably, LoopTool-8B also outperforms the 32B-scale Qwen3 model—used as both the data generator and judge in our pipeline, demonstrating the capability amplification achieved through our model-aware iterative data evolution. On ACEBench (Table 2), LoopTool-8B obtains an overall accuracy of **73.4%**, improving over Qwen3-8B by **+6.3 points** and consistently delivering balanced gains across diverse evaluation categories.

## 5.3 ITERATIVE DETAILS AND ANALYSIS

### 5.3.1 ITERATIVE DATASET DISTRIBUTION

Table 3: Distribution of samples across iterative datasets in our LoopTool framework.

| | # Total | # $\mathcal{D}_j^{\text{ES}}$ | # $\mathcal{D}_j^{\text{EE}}$ | # $\mathcal{D}_j^{\text{HPPL}}$ | # $\mathcal{D}_j^{\text{Seed-new}}$ |
|---|---|---|---|---|---|
| $\mathcal{D}_1$ | 18304 | 0 | 0 | 0 | 18304 (100%) |
| $\mathcal{D}_2$ | 18304 | 1919 (10.48%) | 6566 (35.87%) | 4187 (22.98%) | 5632 (30.77%) |
| $\mathcal{D}_3$ | 18304 | 3386 (18.50%) | 8066 (44.07%) | 4036 (22.06%) | 2816 (15.38%) |
| $\mathcal{D}_4$ | 18304 | 3731 (20.38%) | 8169 (44.63%) | 4996 (27.29%) | 1408 (7.69%) |

The initial seed dataset $\mathcal{D}_{\text{seed}}$ includes $28k$ tool call samples. The corpus $\mathcal{D}_{j+1}$ at iteration $j + 1$ is constructed from four primary sources as illustrated in Eq (5). $\mathcal{D}_j^{\text{Seed-new}}$ means the untrained new seed samples randomly drawn from the seed dataset $\mathcal{D}_{\text{seed}}$. In each iteration, we gradually reduce the proportion of untrained seed samples, ensuring that each training round incorporates

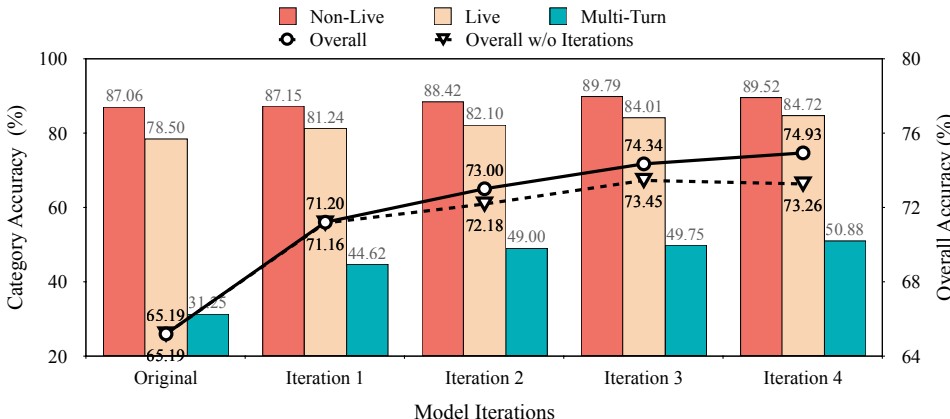

Figure 2: The Iterative Performance across four iterations evaluated in BFCL-v3. The left y-axis represents Category Acc (bar chart), while the right y-axis denotes Overall Acc (line chart)."Overall w/o Iterations" refers to the result obtained under the same number of iteration steps, where we train solely on the initial seed dataset $\mathcal{D}_{\text{seed}}$.

newly generalized queries, while gradually converging on increasingly challenging samples. The detailed data statistics are presented in Table 3.

### 5.3.2 PERFORMANCE ANALYSIS OF ITERATIVE TRAINING FRAMEWORK

We evaluate the effectiveness of the iterative training paradigm against conventional static data generation. As shown in Figure 2, the proposed LoopTool framework delivers consistent gains in tool-calling accuracy across iterations. Starting from the initial model ("Original"), each iteration leverages the closed-loop data evolution to uncover and remedy model deficiencies, leading to steady improvements. In contrast, the static "Overall w/o Iterations" setting produces substantially smaller improvements. Without the injection of newly synthesized hard cases or label refinements, the model rapidly saturates on the limited supervision, exhausting the information content of $\mathcal{D}_{\text{seed}}$. Improvements plateau by Iteration 2 and decline after Iteration 3, indicating overfitting and a growing mismatch between the fixed training distribution and the model's evolving inference behavior. The detailed training curves of multiple iterations of GRPO are presented in Appendix F.

### 5.4 ABLATION STUDY

Table 4: We conduct the corresponding ablation experiments in Iteration 2 and Iteration 3, employing the data variants of $\mathcal{D}_2$ and $\mathcal{D}_3$. Overall accuracy and per-category accuracy are reported.

| Configuration | Overall Acc | Non-Live Acc | Live Acc | Multi-Turn Acc |
|---|---|---|---|---|
| Iteration 1 ($\mathcal{D}_1$) | 71.20 | 87.10 | 81.34 | 44.62 |
| Iteration 2 ($\mathcal{D}_2$) | **73.00** | **88.42** | 82.10 | **49.00** |
|   w/o High-PPL | 72.31 | 88.17 | 81.59 | 46.25 |
|   w/o JGLV | 71.30 | 87.90 | 82.05 | 43.88 |
|   Remove EDDE | 71.50 | 88.06 | 81.47 | 45.00 |
|   HighPPL-Replace | 72.50 | 88.10 | **82.36** | 47.88 |
|   Error-Seed Repetition | 72.38 | 88.40 | 81.87 | 46.88 |
| Iteration 3 ($\mathcal{D}_3$) | **74.34** | **89.79** | **84.01** | **49.75** |
|   w/o High-PPL | 73.50 | 89.12 | 82.79 | 48.90 |
|   w/o JGLV | 72.61 | 89.17 | 82.59 | 46.25 |
|   Remove EDDE | 73.12 | 88.75 | 82.45 | 48.75 |
|   HighPPL-Replace | 73.28 | 89.40 | 83.96 | 46.88 |
|   Error-Seed Repetition | 73.43 | 88.15 | 83.74 | 48.38 |

To assess the contributions of each key component in LoopTool, we perform ablation experiments on BFCL-v3. Specifically, we design the following variants: (i) **w/o High-PPL**: Replace $\mathcal{D}_j^{\text{HPPL}}$ with randomly samples that the model predicted correctly; (ii) **w/o JGLV**: Skip verification and

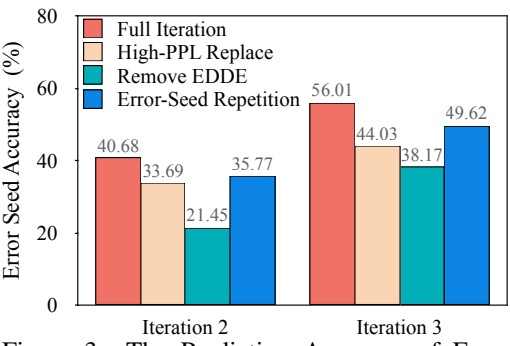

Figure 3: The Prediction Accuracy of Error Seed across iterations.

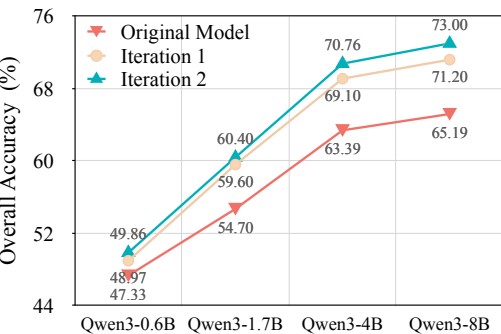

Figure 4: Scaling performance with different model sizes.

treat all mismatches ($a_t \neq a_t^*$) as model errors; keep original labels without refinement. (iii) **Remove EDDE**: Drop $\mathcal{D}_j^{EE}$ entirely; (iv) **HighPPL-Replace**: Replace $\mathcal{D}_j^{EE}$ with an equal number of high-PPL samples selected via GCP; (v) **Error-Seed Repetition**: Remove $\mathcal{D}_j^{EE}$ and duplicate $\mathcal{D}_j^{ES}$ to match data scale. From the results in Table 4, several key observations can be made: From the results in Table 4, several key observations can be made:

- **Importance of high-PPL samples**. **w/o High-PPL** lead to consistent accuracy drops, especially in Multi-Turn cases. Even replacing EDDE samples with high-PPL ones (**HighPPL-Replace**) sustains performance close to full configurations, confirming that high-PPL cases—though previously predicted correctly—lie near decision boundaries of current policy and drive further refinement, in line with recent works (Liang et al., 2025; Shang et al., 2025).
- **Necessity of JGLV.** Skipping verification (**w/o JGLV**) significantly degrades accuracy, confirming that noisy or erroneous labels can mislead training. Without label refinement, such errors persist and even propagate when used by EDDE to generate variants, exacerbating noise in subsequent iterations.
- **Effectiveness of EDDE** The three variants of **w/o EDDE** in both Iteration 2 and Iteration 3, result in consistent drops in overall accuracy. To further quantify the direct contribution of EDDE-originated samples, we compare the three variants with full configuration, testing the accuracy exclusively on this subset of historically wrong cases, with results shown in Figure 3. The result illustrates that simply re-training the model on the original erroneous seeds is insufficient for the model to master these difficult cases effectively. In contrast, EDDE synthesizes structurally similar, error-informed variants that preserve the underlying challenges of the original failure cases while offering additional diversity. This targeted augmentation enables the model to acquire the relevant patterns more reliably, thereby improving its performance on the original hard seeds.

## 5.5 SCALING PERFORMANCE WITH MODEL SIZE

We evaluate LoopTool across backbone models from 0.6B to 8B parameters, measuring BFCL-v3 accuracy over two training iterations (Figure 4). Larger models consistently achieve higher accuracy in both the initial (*Iteration 1*) and refined (*Iteration 2*) stages, with greater absolute improvements in the second iteration. Specifically, the 0.6B model gains only +0.70 points, whereas the 8B model achieves +1.80 points. This scaling trend stems from GRPO-based post-training, which depends on the model's ability to discover correct tool-use trajectories during rollout exploration. Larger models tend to identify such trajectories earlier, thereby amplifying the benefits of iterative refinement.

## 6 CONCLUSION

We propose **LoopTool**, an automated, model-aware framework that unifies data synthesis, label refinement, sample augmentation, and GRPO-based post-training in a closed loop to strengthen tool-augmented LLMs. By using a single open-source model for both generation and judgment, LoopTool produces progressively cleaner and more challenging data without relying on costly closed-source APIs. Our 8B-scale model surpasses even its 32B-scale generator, demonstrating the performance gains enabled by iterative, model-guided data evolution. Future work will investigate online or streaming updates and parallelized iteration to accelerate and further adapt the data–model co-evolution process.

# 7 ETHICS STATEMENT

We acknowledge and adhere to the ICLR Code of Ethics throughout the course of this work. Our study does not involve any human subjects, crowdsourced annotations, or personal data; all datasets used are either publicly available or synthetically generated by the open-source model. To mitigate potential risks of generating harmful or biased content, we adopt standard safety measures in prompt design and apply post-generation filtering on all synthetic data to remove instances that contain toxic, discriminatory, or privacy-violating content.

# 8 REPRODUCIBILITY STATEMENT

We have taken multiple steps to ensure the reproducibility of our results. The full implementation of LoopTool, including the iterative pipeline for model-aware data generation, Greedy Capability Probing, Judgement-Guided Label Verification, and Error-Driven Data Expansion, will be released in an open-source repository upon publication (with an anonymous pre-release link provided for review https://anonymous.4open.science/r/LoopTool). Details on dataset construction, filtering, and augmentation—including prompt templates, generation parameters, and annotation refinement rules—are documented in Appendix.

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

## A  THE USE OF LARGE LANGUAGE MODELS (LLMS)

In the research process, we employed the open-source Language model as both the Judge Model and the data Generator within our proposed LoopTool framework. During manuscript preparation, we used general-purpose LLMs exclusively for grammar checking, phrasing refinement, and clarity improvements in the English text. All conceptual contributions, experiment designs, analyses, and claims in this paper are the responsibility of the authors.

## B  EXPERIMENTAL DETAILS

### B.1  BENCHMARKS

**BFCL** The Berkeley Function-Calling Leaderboard (BFCL-V3) (Patil et al., 2025) constitutes a broad and systematic framework designed to rigorously evaluate the function-calling proficiency of large language models (LLMs) across a diverse spectrum of programming languages, application domains, and intricate real-world scenarios. The benchmark encompasses tasks ranging from multiple and parallel function invocations to multi-turn and multi-step function-call interactions. In total, BFCL-V3 comprises 4,951 test instances—3,951 single-turn cases and 1,000 multi-turn samples-carefully curated to reflect dynamic, authentic use cases. The assessment methodology in BFCL incorporates several complementary metrics:

- **Abstract Syntax Tree (AST) Evaluation**: This metric examines the structural correspondence between the abstract syntax tree of the model-generated output, the ground-truth reference, and the formal function specification. It evaluates the correctness of function identification, the inclusion and accuracy of obligatory parameters, and the precision of both parameter types and associated values.

- **Executable Function Evaluation**: Here, the produced API call is executed, and its runtime output is compared directly against the expected ground-truth result, thereby measuring practical execution accuracy.

- **Multi-turn State-based Evaluation**: The evaluation focus on comparing the backend system's state after all function calls are executed at the end of each turn of the conversation. It capture the correctness of model executions that modify the internal state via write and delete.

- **Multi-turn Response-based Evaluation**: It compares the model's execution path against the minimal viable execution result paths as labeled in ground truth. The minimal viable execution result paths refer to a list of function calls that must be executed in order to produce desired response as user requests.

- **Irrelevance**: This criterion quantifies the model's capacity to avoid generating function calls when presented with extraneous or unrelated user queries. The irrelevance score is determined by dividing the number of accurate non-function-call responses by the total test set size.

- **Relevance**: Relevance gauges the model's adeptness at producing function calls that align contextually with the user's query, irrespective of parameter value accuracy. This score is computed as the proportion of appropriate function-call responses within the entire evaluation set.

**ACEBench** ACEBench (Chen et al., 2025a) is designed to evaluate tool-use capabilities with fine-grained categorization which could be divided into three primary categories: Normal, Special, Agent."Normal" evaluates tool usage in basic scenarios;"Special" evaluates tool usage in situations with ambiguous or incomplete instructions;"Agent" evaluates tool usage through multi-agent interactions to simulate real-world, multi-turn dialogues:

- **Normal Evaluation** compares the model's function call output with the ground truth using AST parsing.

- **Special Evaluation** mainly assesses the ability of model in problem identification. Specifically, the model must: (1) detect and alert missing parameters, (2) accurately locate erroneous parameters, and (3) recognize task-function mismatches.

- **Agent Evaluation** focus on the model's proficiency in utilizing tools during human-agent interactions, employing gpt-4o as a user simulator, incluing End-to-End Accuracy and Process Accuracy.

### B.2 HYPER-PARAMETERS

The detailed hyperparameters of GRPO training are illustrated in Table 5.

## C THE ALGORITHM OF LOOPTOOL

We present the complete procedure of our **LoopTool** framework in Algorithm 1 , which couples *GRPO-based post-training*, *Greedy Capability Probing (GCP)*, *Judgement-Guided Label Verification (JGLV)*, and *Error-Driven Data Expansion (EDDE)* into a unified closed-loop data evolution process.

## D GENERALIZATION ABILITY EVALUATION

Beyond tool-use performance, we evaluate whether the LoopTool-8B model maintains or improves generalization to non-tool-related domains. We compare LoopTool-8B with the vanilla

Table 5: Configuration for Iterative GRPO training.

| Category | Hyperparameter |
|---|---|
| Data Configuration | Train Batch Size: 128 |
| | Validation Batch Size: 128 |
| | Max Prompt Length: 4096 |
| | Max Response Length: 1024 |
| Optimization | Learning Rate: 1e-6 |
| | PPO Mini Batch Size: 128 |
| | KL Loss Used: False |
| | Entropy Loss Used: False |
| | Clip Ratio Low: 0.2 |
| | Clip Ratio High: 0.28 |
| Rollout Configuration | Rollout Mini Batch Size: 2 |
| | GPU Memory Utilization: 0.5 |
| | Number of Rollouts: 12 |
| Training & Logging | Save Frequency (epoch): 1 |
| | Test Frequency (epoch): 1 |

Qwen3-8B (Yang et al., 2025) across six representative benchmarks: MMLU-redux (Gema et al., 2025), IFEVAL (Zhou et al., 2023), LiveCodeBench (Jain et al., 2024), Math-500 (Lightman et al., 2023), AIME24 and AIME25 AIM. The result is illustrated in Table 6.

LoopTool-8B consistently matches or surpasses Qwen3-8B across all domains, with notable improvements in instruction-following (+1.40 on `IFEval`), code generation (+3.84 on `LiveCodeBench`), and mathematics (+1.20 on `Math-500`, and gains on both `AIME` sets. These results indicate that the proposed iterative, model-aware data refinement and training paradigm avoids overfitting to tool-calling tasks. Instead, it fosters improved general reasoning and problem-solving skills, enhancing the model's capacity to generalize across diverse scenarios.

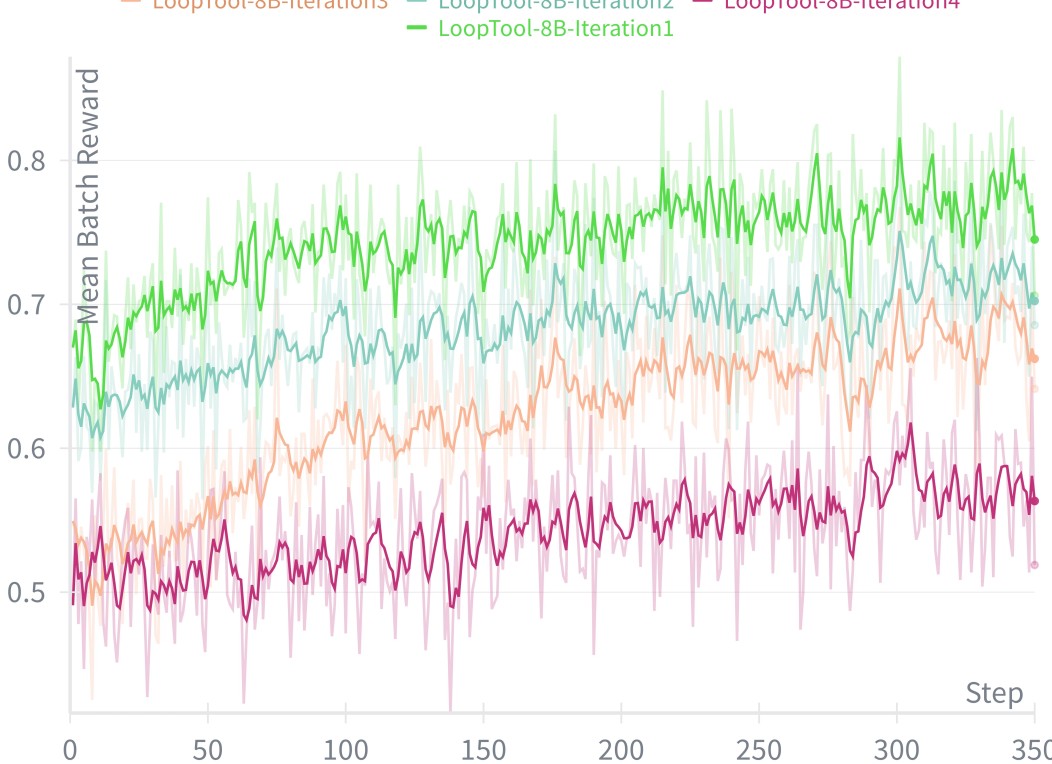

Figure 5: The visualization of reward score across four iterations.

---

**Algorithm 1:** LoopTool: Iterative Model-Aware Data Evolution Framework

---

**Input:** Initial seed dataset $\mathcal{D}_{\text{seed}}$ from Automated Tool-Augmented Data Construction; Initial model parameters $\pi_{\theta_0}$.

**Output:** Final optimized tool-calling model $\pi_{\theta_J}$ after $J$ iterations.

1 **Initialize:** $j \leftarrow 1, \mathcal{D}_1 \leftarrow \text{Subset}(\mathcal{D}_{\text{seed}})$.

2 **while** $j \leq J$ **do**

    `// Step 1:  GRPO-based Post-training`

3    Train policy $\pi_{\theta_{j-1}}$ on $\mathcal{D}_j$ using GRPO in Eq.(2) with binary reward $r(\cdot)$, obtaining updated parameters $\pi_{\theta_j}$.

    `// Step 2:  Greedy Capability Probing (GCP)`

4    **foreach** $(\mathcal{T}, c_t, a_t^*) \in \mathcal{D}_j$ **do**

5        Generate $a_t$ via deterministic greedy decoding from $\pi_{\theta_j}$;

6        **if** $a_t \neq a_t^*$ **then**

7            Send $(\mathcal{T}, c_t, a_t^*; a_t)$ to JGLV for evaluation;

8        Compute $\text{PPL}_{(\mathcal{T}, c_t)}$ by Eq.(3) and retain high-PPL samples and $a_t = a_t^*$ into $\mathcal{D}_j^{HPPL}$;

    `// Step 3:  Judgement-Guided Label Verification (JGLV)`

9    **foreach** *mismatched case* $(\mathcal{T}, c_t, a_t^*; a_t)$ **do**

10       Obtain judgement result
          $y_{\text{judge}} \in \{\text{PRED\_WRONG}, \text{LABEL\_WRONG}, \text{BOTH\_CORRECT}, \text{BOTH\_WRONG}\}$ via Qwen3-32B;

11      **if** $y_{\text{judge}} = PRED\_WRONG$ **then**

12          Add to $\mathcal{D}_j^{MR}$;

13      **else if** $y_{\text{judge}} = LABEL\_WRONG$ **then**

14          Replace $a_t^* \leftarrow a_t$ and add to $\mathcal{D}_j^{LR}$;

15      **else if** $y_{\text{judge}} \in \{BOTH\_CORRECT, BOTH\_WRONG\}$ **then**

16          Discard sample;

    `// Step 4:  Error-Driven Data Expansion (EDDE)`

17    Construct error seed set $\mathcal{D}_j^{ES} \leftarrow \mathcal{D}_j^{MR} \cup \mathcal{D}_j^{LR}$;

18    **foreach** *error seed in* $\mathcal{D}_j^{ES}$ **do**

19        Generate $k$ new samples with scenario diversification constraints;

20    Validate generated set via rule-based + LLM-based evaluation to obtain $\mathcal{D}_j^{EE}$;

    `// Step 5:  Dataset Update for Next Iteration`

21    Select untrained subset $\mathcal{D}_j^{\text{Seed-new}} \subset \mathcal{D}_{\text{seed}}$;

22    Construct next-round dataset by Eq.(5):

$$\mathcal{D}_{j+1} = \mathcal{D}_j^{ES} \cup \mathcal{D}_j^{EE} \cup \mathcal{D}_j^{HPPL} \cup \mathcal{D}_j^{\text{Seed-new}}$$

23    $j \leftarrow j + 1$;

24 **return** $\pi_{\theta_J}$

---

Table 6: Generalization benchmark performance comparison between vanilla Qwen3-8B and LoopTool-8B. Bold indicates the better score for each task.

| Model | MMLU-redux | IFEval | LiveCodeBench | Math-500 | AIME24 | AIME25 |
|---|---|---|---|---|---|---|
| Qwen3-8B | **87.72** | 83.30 | 42.31 | 91.40 | 60.00 | 56.67 |
| LoopTool-8B | 87.37 | **84.70** | **46.15** | **92.60** | **70.00** | **66.67** |

## E  THE PERFORMANCE OF LOOPTOOL ON LLAMA

LoopTool is essentially a **model-agnostic** framework for iterative evolution of data and models. To further verify the effectiveness of the LoopTool framework for other model architectures, we conducted additional experiments using Llama-3.1-8B-Instruct (Grattafiori et al., 2024) as the main

Table 7: Performance comparison of Llama-3.1-8B-Instruct across iterations.

| Model Version | Overall | Non-Live Acc | Live Acc | Multi-Turn Acc | Irrelevance |
|---|---|---|---|---|---|
| Original | 49.72 | 84.48 | 61.13 | 9.62 | 48.46 |
| Iteration-1 | 53.99 | 85.46 | 71.97 | 11.75 | 75.29 |
| Iteration-2 | 56.39 | 86.17 | 74.14 | 14.12 | 82.55 |
| Iteration-3 | 61.00 | 86.73 | 77.74 | 18.13 | 83.93 |

updated policy, while keeping training budget, number of steps, and all LoopTool configurations identical to the Qwen-based runs. We still employ the Qwen3-32B model as both the Generator and the Evaluator. The results are illustrated in the Table 7. Results show clear iterative improvements: overall accuracy rises from 49.72% (Original) to 61.00% (Iteration-3), with substantial metric-level gains such as +8.51 points in Multi-Turn accuracy and +35.37 points in Irrelevance. These results provide direct evidence that the LoopTool framework offers substantial benefits to a different model family (Llama), demonstrating that the improvements are not specific to Qwen-based models.

## F  THE LEARNING CURVES IN ITERATIVE LEARNING

Figure 5 presents a detailed depiction of the reward curves for **LoopTool-8B** across multiple iterative training cycles. These curves consistently exhibit two notable characteristics: (1) **There is a stable increase in reward scores across iterations.** In all four training iterations, the model's average binary reward score curves maintain a consistent upward trajectory, free from oscillations or divergence. This stable improvement mirrors the benchmark results shown in Figure 2, thereby confirming a sustained enhancement in the model's tool-use capability; (2) **The reward trends reflect the escalating difficulty of the data.** As illustrated in Figure 5, the overall data complexity progressively increases from one iteration to the next. Although the absolute reward scores tend to be lower in later iterations due to this heightened difficulty, the curves within each training phase still display a steady upward progression, indicating effective learning despite more challenging conditions.

## G  THE EXAMPLE OF HIERARCHICAL DUAL SUBTREES

The example subtrees of the Context Tree and Constraint Tree are illustrated in Figure 6 and Figure 7, respectively.

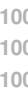
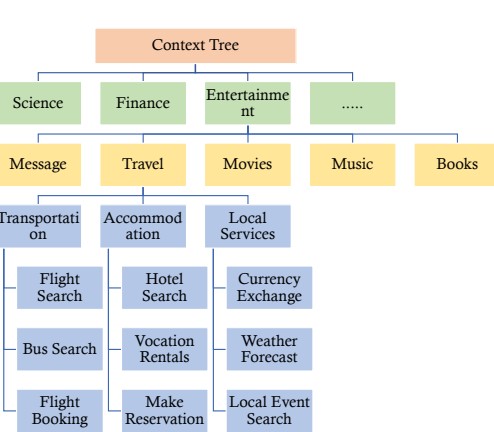

Figure 6: The example subtree of Context Tree.

Figure 7: The example subtree of Constraint Tree.

## H  THE TRAINING SAMPLE FOR GRPO

The Instruction Prompt used in all GRPO samples is illustrated in Figure 8. The Single-Turn and Multi-Turn samples are illustrated in Figure 9 and Figure 10.

## The Full Instruction Content

You are an expert in composing functions. You are given a question and a set of possible functions. Based on the question, you will need to make one or more function/tool calls to achieve the purpose.
If none of the functions can be used, point it out. If the given question lacks the parameters required by the function, also point it out. If the result of tool calls has fulfilled the user's request, summarize the answer.
**Important Notes**
1.  When the tool call has fulfilled the user's request, please provide a. concise summary in plain text without extra tool calls. If no tool is suitable, state that explicitly. If the user's input lacks required parameters, ask for clarification
2.  During each tool invocation, it is important to carefully examine the corresponding tool's description and constraints. Ensure that the required fields of the tool are strictly satisfied, and that parameter types conform to the definitions. If a function call uses the default parameter value, it is not necessary to specify the value during the call.
3.  If the user's request cannot be completed through a one-time function call, or if the parameters of subsequent function calls depend on the results of previous calls, then decompose it into multi-step calls. You only need to return the result of the first step. The use of fictitious parameters or placeholder is strictly prohibited.
4.  In multi-turn dialogs, if you encounter an error and the task remains unfinished, retry with more necessary tool calls until completion. Based on the tool feedback, reflect on whether the understanding or selection of the tool is wrong, what tool calling step is missing, and how to achieve the task goal from now on.
The current time is {current_time}.
# Tools
You may call one or more functions to assist with the user query.
You are provided with function signatures within <tools></tools> XML tags:
<tools>{tool_sets}</tools>
For each function call, return a json object with function name and arguments within <tool_call></tool_call> XML tags:\n<tool_call>\n{\"name\": <function-name>, \"arguments\": <args-json-object>}\n</tool_call>."

Figure 8: The general instruction prompt employed in all GRPO samples. The variables *current_time* and *tool_sets* are placeholders.

## The Single-Turn GRPO Sample

**Input:** **<|im_start|>user\n** I am planning a marketing campaign targeting men within age 25-30 in New York. I need to know the estimated size of my target audience. Also, I want to retrieve the media plan with the Campaign ID 'CAM12345'. Can you help with these? **<|im_end|>**
**Label:** **<tool_call>\n** {"name": "getAudienceSize", "arguments": {"age_range": "25-30", "gender": "male", "location": "New York"}} \n **</tool_call>**
**<tool_call>\n** {"name": "getMediaPlan", "arguments": {"campaign_id": "CAM12345"}} **\n</tool_call>**

Figure 9: The example of Single-Turn GRPO samples.

## The Multi-Turn GRPO Sample

**Input:** **<|im_start|>user\n** I have an image at 'images/satellite_view.png' and I want to understand what the dominant features are in that image. Can you analyze it using a detailed question about its content? **<|im_end|>**
**<|im_start|>assistant\n** **<tool_call>\n**{"name": "pipeline", "arguments": {"image_path": "images/satellite_view.png", "question": "What are the dominant features visible in this satellite view, and how are they distributed across the image?"}}**\n</tool_call>** **<|im_end|>**
**<|im_start|>user\n** **<tool_response>\n**{"caption": "The satellite view reveals a diverse landscape dominated by dense forest regions, intersected by rivers flowing in a northwesterly direction. Urban clusters are sparsely distributed along the riverbanks, with major infrastructure visible in the central part of the image. Open land patches and agricultural fields are scattered towards the southern quadrant."}**\n</tool_response>** **<|im_end|>**
**<|im_start|>assistant\n** The satellite image analysis … **<|im_end|>**
**<|im_start|>user\n** I'd like to calculate the area under the curve for the polynomial function defined by the coefficients [3, -4, 1] between the limits of 1 and 5. Also, can you search for the book 'Calculus: Early Transcendentals' in the library located in New York? **<|im_end|>**

**Label:** **<tool_call>\n** {"name": "mathematics.calculate_area_under_curve", "arguments": {"polynomial": [3, -4, 1], "limits": [1, 5]}} **\n</tool_call>**

**<tool_call>\n** {"name": "library.search_book", "arguments": {"book_name": "Calculus: Early Transcendentals", "city": "New York"}} **\n</tool_call>**

Figure 10: The example of Multi-Turn GRPO samples.

## I  THE LABEL VERIFICATION PROMPT

The Prompt used in Judge-Guide Label Verification (JGLV) is concluded in Figure 11. Sample examples with $y_{judge}$ = PRED_WRONG and $y_{judge}$ = REF_WRONG are respectively presented in Figures 12 and 13.

---

**The Prompt For Judgement-Guided Label Verification (JGLV)**

**System:** You are an expert evaluator specialized in assessing function call responses. Your task is to compare two different function call responses and determine their correctness. Here are the available tools in the conversation:
**<tools> {tool_str}**\n **</tools>**
Evaluation Criteria:
1. Correctness: Whether the function calls properly address the user's request
2. Parameter Accuracy: Whether all parameters are correct and appropriate
3. Function Selection: Whether the chosen functions are suitable for the task
4. Completeness: Whether the response fully satisfies the user's needs.
Please provide objective and thorough evaluations based on these criteria.

**User:** ## **Task**
Please evaluate two function call responses for the following conversation. The time of the conversation is **{date}.**

**Original Conversation:**
**{conversation_text}**

## **Responses to Compare**
Response 1
**{response1}**

Response 2
**{response2}**

## **Output Format**
Strictly respond with the following formats (no additional text):
"[RESPONSE1_INCORRECT/RESPONSE2_INCORRECT/BOTH_CORRECT/BOTH_INCORRECT]
Error Analysis: [Only briefly describe the cause of the incorrect response, without comparing it to the correct response - max 2 sentences] Correct Approach: [Explain the right way to handle this task - max 2 sentences]."

The analysis should be brief and focus on the key differentiating factor.

Figure 11: The Prompt used in Judge-Guide Label Verification for Judgement Model. The red text corresponds to variables that are placeholders.

---

**The PRED_WRONG Example Identified by JGLV**

**Input:** I have a tweet in English that I want to analyze for sentiment. Also, I have a number 1456 that I want to spell out in French. Additionally, I want to check the metadata of my company's website URL and lastly, I need a list of available spell-check rule sets for Dutch. Here is my tweet: "I just love the holidays! Best time of the year!\". And my company's website URL is "https://examplecompany.com".

**Label (Correct):**
**<tool_call>**\n{"name": "Text Sentiment Analysis", "arguments": {"text": "I just love the holidays\"SpellOut\", "arguments": {"data": 1456, "lang": "fr", "ruleset":! Best time of the year!"}}**\n</tool_call>**
**<tool_call>**\n{\"name\": \"SpellOut\", \"arguments\": {\"data\": 1456, \"lang\": \"fr\", \"ruleset\": \"cardinal\"}}**\n</tool_call>**
**<tool_call>**\n{"name": "parse", "arguments": {"url": "https://examplecompany.com"}}**\n</tool_call>**
**<tool_call>**\n{"name": "Get Rule Sets", "arguments": {"lang": "nl"}}**\n</tool_call>**

**Response (Wrong):**
 **<tool_call>**\n{"name": "Text Sentiment Analysis", "arguments": {"text": "I just love the holidays\"SpellOut\", "arguments": {"data": 1456, "lang": "fr", "ruleset":! Best time of the year!"}}**\n</tool_call>**
**<tool_call>**\n{"name": "Get Rule Sets", "arguments": {"lang": "fr"}}**\n</tool_call>**
**<tool_call>**\n{"name": "parse", "arguments": {"url": "https://examplecompany.com"}}**\n</tool_call>**
**<tool_call>**\n{"name": "Get Rule Sets", "arguments": {"lang": "nl"}}**\n</tool_call>**

**Error Message (Generated From Judge Model):**
**Error Analysis:** Response 2 incorrectly uses "Get Rule Sets" with `lang: "fr"` (French) instead of `lang: "nl"` (Dutch) for the Dutch spell-check rule sets request.
**Correct Approach:** Use "SpellOut" with `lang: "fr"` for the French number spelling and "Get Rule Sets" with `lang: "nl"` to retrieve Dutch rule sets as specified.

Figure 12: The example with $y_{judge}$ = PRED_WRONG identified by JGLV.

---

## J  THE ERROR GENERATION PROMPT AND NEW ERROR SAMPLES

The system and user prompts for Error-Driven Data Expansion (EDDE) are illustrated in Figures 14 and 15, respectively. The generated sample case is shown in Figure 16.

## The REF_WRONG Example Identified by JGLV

**Input:** Can you help me calculate the fluctuation in the exchange rate of the Euro against the Dollar between January 1, 2020, and January 1, 2021? Also find the fluctuation for the Pound against the Dollar for the same time period.

**Label (Wrong):**
`<tool_call>\n`{\"name\": \"Fluctuation Endpoint\", \"arguments\": {\"start_string\": \"2020-01-01\", \"end_string\": \"2021-01-01\", \"base\": \"usd\", \"symbols\": \"eur\"}}`\n</tool_call>`
`<tool_call>\n`{\"name\": \"Fluctuation Endpoint\", \"arguments\": {\"start_string\": \"2020-01-01\", \"end_string\": \"2021-01-01\", \"base\": \"usd\", \"symbols\": \"gbp\"}}`\n</tool_call>`

**Response (Correct):**
`<tool_call>\n`{\"name\": \"Fluctuation Endpoint\", \"arguments\": {\"start_string\": \"2020-01-01\", \"end_string\": \"2021-01-01\", \"base\": \"eur\", \"symbols\": \"usd\"}}`\n</tool_call>`
`<tool_call>\n`{\"name\": \"Fluctuation Endpoint\", \"arguments\": {\"start_string\": \"2020-01-01\", \"end_string\": \"2021-01-01\", \"base\": \"gbp\", \"symbols\": \"usd\"}}`\n</tool_call>`

**Error Message (Generated From Judge Model):**
**Error Analysis:** Response 1 incorrectly sets USD as the base currency and EUR/GBP as symbols, which calculates USD's fluctuation against EUR/GBP instead of EUR/GBP's fluctuation against USD. The base and symbols parameters were reversed.

**Correct Approach:** Use EUR and GBP as the base currencies with USD as the symbol in separate calls to measure their respective fluctuations against the Dollar.

Figure 13: The example with $y_{judge} = $ REF_WRONG identified by JGLV.

## The System Prompt for Error-Driven Data Expansion (EDDE)

**System:** You are an expert data engineer specialized in creating high-quality training samples for tool-calling language models. Your mission is to generate NEW training samples that help models learn correct tool usage patterns.
## Context Information

**Current Date**: {current_date}
**Available Tools**:
`<tools>` {tools_content} `</tools>`

## Your Task
You will analyze an error case and create a completely NEW sample that:

1. Demonstrates CORRECT tool usage in a similar scenario
2. Uses different parameters (names, amounts, symbols, etc.)
3. Maintains similar complexity and potential error patterns
4. Follows exact format requirements

## Critical Requirements

1. Generate a realistic but DIFFERENT scenario;
2. Show the CORRECT approach to handle such requests;
3. Use proper conversation format with role markers.
4. Output format must be: INPUT: [content] \n OUTPUT: [content];
5. INPUT could be a single turn or multiple turn request; OUTPUT must use `<tool_call>\n...\n</tool_call>` format
6. NO additional explanations or text outside the INPUT/OUTPUT sections

Figure 14: The system prompt for Error-Driven Data Expansion (EDDE).

## The User Prompt for Error-Driven Data Expansion (EDDE)

**User:** According to the original sample, generate a NEW training sample:

**Original Sample**: {converted_input}
**Correct Response**: {correct_response}
**Incorrect Response**: {incorrect_response}
**Error Analysis:** {error_analysis}

**Format Requirements**
Your response must ONLY contain:
INPUT: [New conversation with proper role markers like USER, ASSISTANT]
OUTPUT: [Correct tool calls in `<tool_call>\n...\n</tool_call>` format]

**Generation Instructions**
Create a NEW sample that addresses the same type of error but with maximum diversity:
- Apply the scenario focus: {scenario['focus']}
- Follow parameter strategy: {scenario['parameters']}
- The new sample should help the model recognize the error pattern from the original case and learn the correct approach for similar scenarios
- The correct invocation of new samples does not need to use exactly the same tools as the original samples, and can further involve more complex user requests.

**Format Requirements**

Your response must ONLY contain:

INPUT: [New conversation with proper role markers like USER, ASSISTANT]
OUTPUT: [Correct tool calls in `<tool_call>\n...\n</tool_call>` format]

**Some Output Examples:**

{sample_1}

{sample_2}
**...**

Figure 15: The user prompt for Error-Driven Data Expansion (EDDE).

---

**The Generated Single-Turn GRPO Sample from EDDE**

**Original Sample**

**Input:** Log me in with username \"Customer Support\" and password \"P@ssw0rd2023\". Then, show me all tickets I've created

**Label (Correct):** **<tool_call>\n**{"name": "ticket_login", "arguments": {"username": "Customer Support", "password": "P@ssw0rd2023"}}**\n</tool_call>\n<tool_call>\n**{"name": "get_user_tickets", "arguments": {}}**\n</tool_call>"**

**Response (Wrong):** **<tool_call>\n**{"name": "ticket_login", "arguments": {"username": "Customer Support", "password": "P@ssw0rd2023\"}} **\n</tool_call>**

**New Generated Sample**

**Input:** Please sign me in with username "jane.doe" and password "Secure#2024". After that, show me any tickets I have that are currently marked as "open".

**Label:** **<tool_call>\n**{"name": "ticket_login", "arguments": {"username": "jane.doe", "password": "Secure#2024"}}**\n</tool_call>\n<tool_call>\n**{"name": "get_user_tickets", "arguments": {"status": "open"}}**\n</tool_call>**

Figure 16: The new sample generated by EDDE according to the error in the model response.

---

