# OpenReview forum: "LoopTool: Closing the Data–Training Loop for Robust LLM Tool Calls"
_ICLR.cc/2026/Conference — ICLR 2026 Conference Withdrawn Submission_

### Official Review · Reviewer_F8Dd · 2025-10-21

**Soundness:** 2
**Presentation:** 2
**Contribution:** 2
**Rating:** 2
**Confidence:** 4

**Summary:**

This paper introduces LoopTool, a closed-loop, model-aware data evolution framework for improving large language models’ (LLMs) tool-use capabilities. The framework iteratively refines both data and model parameters using GRPO reinforcement learning, achieving significant improvements. The resulting 8B model surpasses its 32B generator and attains state-of-the-art results on BFCL-v3 and ACEBench benchmarks.

**Strengths:**

1. Comprehensive framework design. The integration of GCP, JGLV, and EDDE forms a conceptually complete pipeline that addresses diagnosis, label refinement, and data augmentation in a unified framework.
2. Empirical completeness. The authors present quantitative results on multiple public benchmarks (BFCL-v3 and ACEBench) and include ablations for each component (Table 4), demonstrating systematic experimentation.

**Weaknesses:**

1. Poor presentation quality.
   - The contribution list reads more like an implementation description, especially for contribution 2 and 3.
   - Figures and tables are poorly formatted—many are embedded directly within paragraphs with insufficient spacing, which severely disrupts readability.
   - Figure 1 is visually unpolished, and the label “Greedy Capacity Probing” appears to be a typographical error, as “Capability” is used elsewhere.

2. lack of novalty. The paper claims that existing approaches treat data generation and model training as two non-interactive processes. However, similar ideas have been explored in prior self-adaptation [1] and self-challenging [2] paradigms, where models iteratively generate or select new data based on their own behavior to improve subsequent training. As such, the conceptual contribution of LoopTool appears incremental rather than fundamentally novel.

3. Marginal performance improvements. Despite the complexity of the proposed iterative framework, the reported gains over training solely on the initial seed dataset are relatively modest (see Figure 2). Considering the substantial additional overhead—such as repeated data synthesis and the reliance on a larger model for data generation—the cost–benefit ratio appears unfavorable.

4. Limited experimental scope on model backbone.
The experiments are conducted solely on the Qwen3 series, without evaluation on models of different architectures. As a result, the generality of the proposed approach remains unverified.

[1] A. Zweiger et al. Self-Adapting Language Models

[2] Y. Zhou et al., Self-Challenging Language Model Agents

**Questions:**

1. Given the limited improvement, what is the practical efficiency gain (if any) when accounting for computation and time cost?
2. How does LoopTool compare against simpler iterative fine-tuning baselines that periodically regenerate part of data without the additional JGLV/EDDE modules?

---

> ### Author Response · Authors · 2025-11-21
> **Response to Weakness 1**
>
> > Poor presentation quality.
> >
> > - The contribution list reads more like an implementation description, especially for contributions 2 and 3.
> >
> > - Figures and tables are poorly formatted—many are embedded directly within paragraphs with insufficient spacing, which severely disrupts readability.
> >
> > - Figure 1 is visually unpolished, and the label “Greedy Capacity Probing” appears to be a typographical error, as “Capability” is used elsewhere.
>
> 1. In the revised manuscript, we consolidate and refine Contribution 2 and Contribution 3 from the original paper. The complete and revised version of this list of contributions is presented as follows.
>    - We present **LoopTool**, the first fully automated, model‑aware iterative framework that tightly couples reinforcement learning post‑training with targeted data synthesis. By continuously diagnosing model weaknesses and generating capability‑targeted training data, LoopTool enables dynamic **co‑adaptation** of the model and the dataset.
>    - We propose a closed‑loop data refinement and augmentation strategy that purifies labels through comparative judgment (**JGLV**) and transforms verified failures into diverse, high‑value training samples (**EDDE**), enhancing tool‑use learning without reliance on closed‑source models.
>    - Leveraging fully open-source, self-contained data generation and refinement, our 8B model trained by LoopTool surpasses its 32B generator and achieves state-of-the-art performance on BFCL‑v3 and ACEBench among models of similar scale.
>
> 2. We acknowledge this formatting issue. In the revised version, we have restructured all figures and tables into proper floating environments and added clear spacing above and below each to separate them from the main text. These adjustments improve visual clarity and overall readability.
> 3. In the revision, we correct the terminology in Figure 1 to "**Greedy Capability Probing**", consistent with usage throughout the manuscript.

---

> ### Author Response · Authors · 2025-11-21
> **Response to Weakness 2**
>
> > Weakness 2:  Lack of novelty. The paper claims that existing approaches treat data generation and model training as two non-interactive processes. However, similar ideas have been explored in prior self-adaptation [1] and self-challenging [2] paradigms, where models iteratively generate or select new data based on their own behavior to improve subsequent training. As such, the conceptual contribution of LoopTool appears incremental rather than fundamentally novel.
> >
> > [1] A. Zweiger et al. Self-Adapting Language Models
> >
> > [2] Y. Zhou et al., Self-Challenging Language Model Agents
>
> To address this concern about novelty thoroughly, we compare LoopTool with SEAL [1] and SCA [2] from **three specific perspectives**:
>
> 1. **Contemporaneous Work**. According to the ICLR 2026 [official reviewing policy](https://iclr.cc/Conferences/2026/ReviewerGuide): "We consider papers contemporaneous if they are published within the last two months. That means, since our full paper deadline is September 24, if a paper was published (i.e., at a peer-reviewed venue) on or after July 24, 2025, authors are not required to compare their own work to that paper… arXiv is not considered a peer-reviewed venue…" SEAL [1] was released in June 2025 on arXiv, and SCA [2] was published at NeurIPS 2025 in September 2025, respectively. Our work was fully developed and submitted before the September 24, 2025, deadline, independently of these publications. These works, therefore, represent contemporaneous efforts rather than established approaches upon which LoopTool builds.
>
> 2. **Fundamental Differences in Loop Structure and Data Lifecycle**. SEAL implements an *instance-level self-edit loop*, where synthetic finetuning data is created and consumed in a single adaptation step. No persistent dataset is maintained, and no iterative multi-generation refinement occurs. SCA focuses on *task challenger–executor interaction* in tool-use environments, generating “task definitions” to improve RL/SFT performance. This process produces evaluable tasks rather than large-scale supervised datasets. LoopTool is explicitly **dataset-centric**: (a) Diagnoses capability gaps via GCP, (b) Corrects noisy labels via JGLV that leverages the evolving trainee model’s outputs, (c) Generates diversified, structurally analogous hard cases via EDDE. This structured, multi-module pipeline persistently evolves the training corpus over many iterations—going beyond per-step adaptation or task generation.
>
> 3. **Difference in Objectives and Outcomes**. SEAL optimizes immediate weight updates for knowledge incorporation/few-shot reasoning, with the loop tied directly to adaptation efficiency. SCA discovers challenging tasks to improve RL-based multi-turn tool-use performance in exploratory environments. In contrast, LoopTool builds progressively refined, benchmark-scale tool-use datasets without closed-source APIs, enabling a smaller model (8B) to outperform its own larger data generator (32B) on BFCL-v3 and ACEBench.
>
> In summary, while SEAL and SCA share the broad “iterative training” philosophy, **they were developed in parallel with our work and differ fundamentally in loop granularity, data lifecycle, and objectives**. LoopTool’s novelty lies in operationalizing a cost-effective, open-source, multi-module closed loop for persistent dataset refinement, resulting in measurable capability gains that exceed its generator model.
>
> [1] Self-Adapting Language Models, arxiv 2025
>
> [2] Self-Challenging Language Model Agents, NeurIPS 2025

---

> ### Author Response · Authors · 2025-11-21
> **Response to Weakness 3 & Question 1**
>
> > Weakness 3: Marginal performance improvements. Despite the complexity of the proposed iterative framework, the reported gains over training solely on the initial seed dataset are relatively modest (see Figure 2). Considering the substantial additional overhead—such as repeated data synthesis and the reliance on a larger model for data generation—the cost–benefit ratio appears unfavorable.
>
> > Question 1: Given the limited improvement, what is the practical efficiency gain (if any) when accounting for computation and time cost?
>
> We appreciate the reviewer’s concern regarding the cost–benefit trade‑off of our iterative framework. We address this in three parts:
>
> 1. **The significance of the reported gains.** In the BFCL‑v3 leaderboard (Refer to Table 1 in the paper), the top entries are separated by minimal margins—often less than **1 percentage point** in overall accuracy. Within this context, **Iteration 4** of LoopTool‑8B achieves a +3.73 improvement in overall accuracy over **Iteration 1** (from 74.93 to 78.66), resulting in consistent gains across all major categories: **Non‑Live AST** (+2.37), **Live Execution** (+3.48), and **Multi‑Turn** (+6.26). Such an improvement is substantially larger than the gaps typically observed among top-ranked models, and would correspond to a leap of multiple leaderboard positions. This magnitude of gain thus reflects not only a strong absolute performance boost but also a demonstrable advancement in solving hundreds of additional cases across BFCL‑v3’s most challenging evaluation dimensions.
>
> 2. **The computational overhead of training and data iteration.** We have quantified actual GPU‑hour usage for data iteration and GRPO training, with all experiments run on the same 8 NVIDIA H800 hardware. We deploy the Qwen3-32B model service locally via vLLM to accomplish the JGLV and EDDE modules. The computational cost is equivalently converted into comparable GPU Hours. The result is shown in the table below.
>
>    |             | GCP  | JGLV | EDDE | GRPO Training |
>    | ----------- | ---- | ---- | ---- | ------------- |
>    | Iteration-1 | 0    | 0    | 0    | 173.81        |
>    | Iteration-2 | 8.42 | 2.83 | 6.49 | 171.96        |
>    | Iteration-3 | 8.77 | 3.15 | 7.97 | 181.07        |
>    | Iteration-4 | 9.10 | 3.49 | 8.07 | 188.45        |
>
> **This shows that the per‑iteration data evolution cost is only ~10% of the GRPO training cost**. In contrast, without LoopTool’s closed‑loop data refinement, static‑seed training rapidly converges and begins to overfit (as seen by "Overall w/o Iterations" in Figure 2), leaving additional compute under‑utilized.
>
> 3. **The avoidance of full‑dataset re‑generation.** LoopTool **DOSE NOT** require regenerating the entire dataset per iteration. JGLV verifies only mismatches (≈20% of samples, ≈4k samples), replacing incorrect labels without large‑scale regeneration. EDDE expands only verified “hard” error seeds (≈10–20% of samples, ≈2k–4k samples), each producing 4 diversified variants. As a result, over 60% of each iteration’s training data is reused, and the synthesis cost per round is a small fraction of full‑dataset construction.

---

> ### Author Response · Authors · 2025-11-21
> **Response to Weakness 4**
>
> > Weakness 4: Limited experimental scope on model backbone. The experiments are conducted solely on the Qwen3 series, without evaluation on models of different architectures. As a result, the generality of the proposed approach remains unverified.
>
> To directly address the concern of generality, we conducted additional experiments with **Llama‑3.1‑8B‑Instruct** as the student backbone under identical budgets, training steps, and LoopTool configurations. The Llama‑3.1 directly outputs Python‑style tool calls.
>
> The results in the table below demonstrate **consistent iterative gains**: Overall accuracy rises from 49.72% (Original) to a peak of 61.00% (Iteration‑3), with steady improvements across Non‑Live, Live, Multi‑Turn, and Irrelevance metrics. For example, Multi‑Turn accuracy increases by +8.51 points and Irrelevance by +35.37 points over the original. Performance at Iteration‑4 (60.82%) is comparable to Iteration‑3, indicating that the model has largely saturated and that further iterations yield diminishing returns. These outcomes confirm that **LoopTool’s closed‑loop refinement is effective across architectures and output modalities.**
>
> | Llama-3.1-8B-Instruct | Overall | Non-Live Acc | Live Acc | Multi-Turn Acc | Irrelevance |
> | :-------------------: | :-----: | :----------: | :------: | :------------: | :---------: |
> |       Original        |  49.72  |    84.48     |  61.13   |      9.62      |    48.46    |
> |      Iteration-1      |  53.99  |    85.46     |  71.97   |     11.75      |    75.29    |
> |      Iteration-2      |  56.39  |    86.17     |  74.14   |     14.12      |    82.55    |
> |      Iteration-3      |  61.00  |    86.73     |  77.74   |     18.13      |    83.83    |
> |      Iteration-4      |  60.32  |    87.04     |  75.66   |     17.50      |    85.78    |

---

> ### Author Response · Authors · 2025-11-21
> **Response to Question 2**
>
> > Question2: How does LoopTool compare against simpler iterative fine-tuning baselines that periodically regenerate part of the data without the additional JGLV/EDDE modules?
>
> We thank the Reviewer for this valuable question. In our study, such simpler baselines correspond directly to the **w/o JGLV** and **Remove EDDE** ablation settings in Table 4 — these variants refresh part of the dataset in the loop but omit either label verification or targeted augmentation. Additionally, we supplement an extra baseline that **randomly regenerate an equal number of new samples (significantly consumes more computational resources)** to strictly match the data scale of the full configuration. The corresponding experimental results are rearranged in the table below to facilitate review.
>
> | Configuration                   | Overall Acc | Non-Live Acc | Live Acc | Multi-Turn Acc |
> | ------------------------------- | ----------- | ------------ | -------- | -------------- |
> | Iteration 2                     | 73.00       | 88.42        | 82.10    | 49.00          |
> | Iteration 2 (w/o JGLV)          | 71.30       | 87.90        | 82.05    | 43.88          |
> | Iteration 2 (Remove EDDE)       | 71.50       | 88.06        | 81.47    | 45.00          |
> | Iteration 2 (Random Regeration) | 71.64       | 88.19        | 81.95    | 44.25          |
> | Iteration 3                     | 74.34       | 89.79        | 84.01    | 49.75          |
> | Iteration 3 (w/o JGLV)          | 72.61       | 89.17        | 82.59    | 46.25          |
> | Iteration 3 (Remove EDDE)       | 73.12       | 88.75        | 82.45    | 48.75          |
> | Iteration 3 (Random Regeration) | 72.96       | 88.82        | 82.67    | 47.55          |
>
> Results show that all these simpler iterative refresh strategies perform consistently worse than the full LoopTool configuration. In **Iteration 2**, removing JGLV reduces overall accuracy from 73.00 to 71.30 and multi-turn accuracy from 49.00 to 43.88, while removing EDDE yields a similar drop to 71.50 overall and 45.00 multi-turn. Even the **Random Regeneration** baseline, which regenerates an equal number of samples to match data scale but at much higher computational cost, attains only 71.64 overall and 44.25 multi-turn. The performance gap stems from two factors: **without JGLV**, noisy synthetic labels remain uncorrected and continue to mislead training, and **without EDDE**, the model fails to receive structurally similar, scenario-diversified variants of its own failure cases that are crucial for mastering difficult tool-use scenarios. As confirmed by our analysis in Fig. 3, EDDE-originated samples produce marked improvements on historically hard seeds that random regeneration cannot replicate. **These results demonstrate that LoopTool’s closed-loop, model-aware data refinement delivers superior accuracy and efficiency compared to simpler, label-unchecked regeneration strategies.**

---

> ### Author Response · Authors · 2025-11-26
> **Request for Further Feedback on Rebuttal**
>
> Dear Reviewer F8Dd,
>
> I hope this message finds you well. I have submitted detailed responses to your concerns during the rebuttal phase, and I was wondering if you have any further feedback or if there are any remaining questions regarding my submission. I would be more than happy to provide additional clarifications or address any further points you may have. I completely understand the busy review schedule, and I truly appreciate your time and effort in evaluating my work.

---

> ### Comment · Reviewer_F8Dd · 2025-11-27
>
> Thanks for the authors’ detailed responses, but I still have several concerns.
>
> Regarding novelty, I understand the distinction between this work and prior methods such as SEAL and SCA. However, I am not convinced that looped training itself constitutes a fresh idea. In industry, many models are trained in multiple rounds with targeted data for each stage, which is largely an engineering practice rather than a conceptual innovation. The novelty of this work appears to lie in automating the looped training process, but the effectiveness of this automation remains uncertain.
>
> The authors states that “Iteration of LoopTool-8B achieves a +3.73 improvement in overall accuracy over Iteration 1 (from 74.93 to 78.66).” However, comparing the model after one training iteration to the untrained model is not particularly informative. Based on the curve in Figure 2, when comparing LoopTool with the “Overall w/o Iterations” baseline, the improvements are quite small: only +0.04 in Iteration 1, +0.82 in Iteration 2, +0.89 in Iteration 3, and +1.67 in Iteration 4. Moreover, the “Overall w/o Iterations” baseline is only a simple multi-epoch training using the same data, which is not a strong comparator. Given this, I do not find the reported improvements to be substantial.
>
> Additionally, LoopTool incorporates a much stronger model—Qwen3-32B—within its training loop, which in some sense resembles a form of distillation. Why not simply perform direct distillation instead of employing LoopTool? Alternatively, could additional distillation components be integrated into LoopTool to further improve performance?

---

> > ### Author Response · Authors · 2025-11-30
> > **Response to New Concerns (Part 1)**
> >
> > > Regarding novelty, I understand the distinction between this work and prior methods such as SEAL and SCA. However, I am not convinced that looped training itself constitutes a fresh idea. In industry, many models are trained in multiple rounds with targeted data for each stage, which is largely an engineering practice rather than a conceptual innovation. The novelty of this work appears to lie in automating the looped training process, but the effectiveness of this automation remains uncertain.
> >
> > We acknowledge the reviewer’s observation regarding the existence of multi‑round targeted data training in industry practice. **Our contribution goes beyond mere “looping” of training stages. To our knowledge, LoopTool is the first framework that integrates a model‑aware iterative data evolution loop with GRPO reinforcement learning specifically for toocalling.**
> >
> > (1) **Conceptual novelty from coupling looped training with GRPO for tool‑calling capabilities**
> >
> > Unlike standard multi‑round fine‑tuning—which merely stages static datasets—LoopTool integrates the training loop deeply into the data generation/refinement process, driven by three synergistic modules:
> >
> > - **GCP targets the most informative samples at the model’s capability boundary.** GCP identifies mastered, borderline, and failed tool‑calling cases using greedy decoding and difficulty scoring via perplexity. This directly addresses a unique GRPO limitation: grouped advantage degeneration to near zero for overly simple or overly hard samples, which halts gradient updates. GCP supplies targeted, high‑PPL boundary cases to sustain meaningful GRPO optimization.
> >
> > - **EDDE transforms real failure cases into diverse, challenging variants that directly address model weaknesses.** For verified failures, it generates structurally similar but contextually diversified hard cases, preserving the functional challenge while increasing scenario coverage. Ablation results in Table 4 and Figure 3 show that EDDE‑originated hard samples yield measurable gains (e.g., +5–7 points in error samples) beyond simply repeating original error seeds.
> >
> > - **JGLV continuously purifies supervision by replacing incorrect labels with superior model predictions.** JGLV leverages the synergy between the evolving student policy and a high‑capacity judge model to perform comparative judgement (PRED_WRONG, LABEL_WRONG). This comparative approach is computationally cheaper, avoids full regeneration, and directly integrates into the LoopTool cycle.
> >
> > (2) **Automation effectiveness and computational cost**
> >
> > We have quantified the cost of per‑iteration data evolution (including GCP filtering, JGLV verification, and EDDE generation) relative to GRPO training cost: **as shown in our response to Weakness 3 & Question 1, data evolution costs ~10% of a single GRPO iteration’s GPU hours.** This efficiency is achieved by:
> >
> > - Using a single open‑source model as both generator and judge (Qwen3‑32B);
> >
> > - Updating only subsets of data per iteration rather than regenerating full datasets;
> >
> > Given the observed accuracy gains across iterations (Figure 2), this cost is highly acceptable compared to the benefit delivered and is significantly lower than typical industrial multi‑round practices requiring repeated full‑scale data regeneration.

---

> > > ### Author Response · Authors · 2025-11-30
> > > **Response to New Concerns (Part 2)**
> > >
> > > > The authors state that “Iteration of LoopTool-8B achieves a +3.73 improvement in overall accuracy over Iteration 1 (from 74.93 to 78.66).” However, comparing the model after one training iteration to the untrained model is not particularly informative. Based on the curve in Figure 2, when comparing LoopTool with the “Overall w/o Iterations” baseline, the improvements are quite small: only +0.04 in Iteration 1, +0.82 in Iteration 2, +0.89 in Iteration 3, and +1.67 in Iteration 4. Moreover, the “Overall w/o Iterations” baseline is only a simple multi-epoch training using the same data, which is not a strong comparator. Given this, I do not find the reported improvements to be substantial.
> > >
> > > We believe there is a misunderstanding of the experimental setup and intention behind the “Overall w/o Iterations” baseline and the interpretation of improvements. We clarify as follows:
> > >
> > >
> > >
> > > - **Baseline scale mismatch.** The “Overall w/o Iterations” baseline is not a same-scale comparison to each iteration of LoopTool. In this baseline, the model is trained for multiple epochs on the entire 28k seed dataset repeatedly. By contrast, in each LoopTool iteration, the model trains on only ~18.3k samples, which is substantially smaller. This suggests that the data scale corresponding to “Overall w/o Iterations” is greater than that of a single iteration. Iteration 1 for both methods starts from the **same initialization** (pretrained Qwen3‑8B) and uses the same number of optimization steps. Performance proximity in Iteration 1 is therefore expected.
> > >
> > > - **Gains grow across later iterations.** The key difference emerges in later iterations when LoopTool continually injects refined and challenging samples. Indeed, as noted, the gap grows across iterations: +0.82 (Iteration 2), +0.89 (Iteration 3), and +1.67 (Iteration 4). This trend demonstrates that the benefit of LoopTool is not a one‑off improvement but amplifies over time, as each round leverages model‑specific weaknesses to generate high‑value supervision. In contrast, the static “Overall w/o Iterations” baseline cannot accumulate such iterative benefits, and its performance plateaus and even declines in Iteration 4 due to overfitting on a fixed dataset.
> > >
> > > - **Significance of absolute margins.** In BFCL‑v3, leading models often differ by <1% overall accuracy; thus, gains of +0.8–1.6 points represent observable leaderboard advancements over stronger or even larger‑scale models. From Iteration 1 to Iteration 2, LoopTool surpasses GPT‑4o‑2024‑11‑20 and xLAM‑2‑8B‑fc‑r in overall accuracy. From Iteration 2 to Iteration 3, it further surpasses watt‑tool‑70B.
> > >
> > > In summary, it is important to consider that the seed dataset construction is itself part of LoopTool’s core design. Evaluating the entire process from the original Qwen3‑8B to LoopTool‑8B, the improvement is close to 10% overall accuracy on BFCL‑v3. A similar >10% gain is observed when applying LoopTool to Llama‑3.1‑8B‑Instruct. These consistent, architecture‑independent gains provide strong evidence that LoopTool’s iterative data evolution framework and higher‑quality supervision fundamentally improve tool‑use capability.

---

> > ### Author Response · Authors · 2025-11-30
> > **Response to New Concerns (Part 3)**
> >
> > > Additionally, LoopTool incorporates a much stronger model—Qwen3-32B—within its training loop, which in some sense resembles a form of distillation. Why not simply perform direct distillation instead of employing LoopTool? Alternatively, could additional distillation components be integrated into LoopTool to further improve performance?
> >
> > We address the reviewer’s question about the use of Qwen3‑32B in LoopTool and the suggestion of direct distillation as follows:
> >
> > - **LoopTool is model‑agnostic**. The generator and evaluator in LoopTool’s closed loop can be **any** strong LLM, not limited to Qwen3‑32B. In our main experiments, we intentionally used Qwen3‑32B as the generator/judge and Qwen3‑8B as the target model to show that LoopTool’s iterative process can produce a student model that outperforms the original data generator itself. This setting was selected to emphasize **capability amplification**, not to tie the method to a specific model pair.
> >
> > - **Regarding direct distillation. Conventional one‑shot or static distillation typically transfers the teacher’s outputs as‑is, making it difficult for the student to surpass the teacher’s performance.** In tool‑use scenarios, this risk is amplified: static teacher outputs may contain annotation noise, be suboptimal for certain queries, and cannot adapt to the student’s evolving weaknesses. LoopTool differs fundamentally by tightly coupling training with data evolution. This closed‑loop, iterative **co‑adaptation** of model and data enables the training signal to evolve with the student’s competence — a property absent in distillation.
> >
> > - **On integrating distillation into LoopTool**. Due to time constraints, we did not explore this hybrid in the current work, but we consider it a promising future direction to further validate complementarity between distillation and iterative data‑model co‑evolution.

---

### Official Review · Reviewer_N4cR · 2025-10-23

**Soundness:** 3
**Presentation:** 3
**Contribution:** 3
**Rating:** 6
**Confidence:** 3

**Summary:**

LoopTool is a closed-loop, model-aware data–training system for tool-augmented LLMs. It repeatedly diagnoses capability gaps, verifies/corrects labels, and expands hard examples, then retrains with GRPO—so the data distribution adapts to the model’s evolving needs. On BFCL-v3 and ACEBench, the 8B student surpasses peer open-source models of similar size and, on several dimensions, even outperforms its 32B generator/judge. Ablations and iteration curves show that high-PPL sampling, JGLV, and EDDE each contribute substantially to the gains.

**Strengths:**

1. A fully automatic, model-aware iterative pipeline that tightly couples data generation with training for tool use; continual diagnosis and error-targeted synthesis keep supervision aligned with the model’s evolving capabilities.

2. JGLV and EDDE are well-motivated and practically effective.

3. Solid coverage of executable benchmarks (BFCL-v3, ACEBench) plus thorough ablations and iteration analyses.

**Weaknesses:**

1. Training and closed-loop verification are largely confined to the Qwen family (student, generator, and judge), raising concerns about same-source bias and cross-backbone generalization.

2. The GRPO + binary reward setup lacks convergence and stability analysis.

**Questions:**

1. Does LoopTool generalize beyond Qwen backbones under matched budgets and training steps?

2. Can the RL component provide convergence/stability guarantees or empirical bounds?

---

> ### Author Response · Authors · 2025-11-21
> **Response to Weakness 1& Question 1**
>
> > Weakness 1: Does LoopTool generalize beyond Qwen backbones under matched budgets and training steps? Training and closed-loop verification are largely confined to the Qwen family (student, generator, and judge), raising concerns about same-source bias and cross-backbone generalization.
>
> > Quesion 1: Does LoopTool generalize beyond Qwen backbones under matched budgets and training steps?
>
> To directly address the concern of same‑source bias and cross-backbone generalization, we conducted additional experiments with Llama‑3.1‑8B‑Instruct as the student backbone under identical budgets, training steps, and LoopTool configurations. The Llama‑3.1 directly outputs Python‑style tool calls.
>
> The results in the table below demonstrate **consistent iterative gains**: Overall accuracy rises from 49.72% (Original) to a peak of 61.00% (Iteration‑3), with steady improvements across Non‑Live, Live, Multi‑Turn, and Irrelevance metrics. For example, Multi‑Turn accuracy increases by +8.51 points and Irrelevance by +35.37 points over the original. Performance at Iteration‑4 (60.82%) is comparable to Iteration‑3, indicating that the model has largely saturated and that further iterations yield diminishing returns. These outcomes confirm that **LoopTool’s closed‑loop refinement is effective across architectures and output modalities, and the gains are not due to the same‑source bias.**
>
> | Llama-3.1-8B-Instruct | Overall | Non-Live Acc | Live Acc | Multi-Turn Acc | Irrelevance |
> | :-------------------: | :-----: | :----------: | :------: | :------------: | :---------: |
> |       Original        |  49.72  |    84.48     |  61.13   |      9.62      |    48.46    |
> |      Iteration-1      |  53.99  |    85.46     |  71.97   |     11.75      |    75.29    |
> |      Iteration-2      |  56.39  |    86.17     |  74.14   |     14.12      |    82.55    |
> |      Iteration-3      |  61.00  |    86.73     |  77.74   |     18.13      |    83.83    |
> |      Iteration-4      |  60.32  |    87.04     |  75.66   |     17.50      |    85.78    |

---

> ### Author Response · Authors · 2025-11-21
> **Response to Weakness 2 & Question 2**
>
> > Weakness 2: The GRPO + binary reward setup lacks convergence and stability analysis.
>
> > Question 2: Can the RL component provide convergence/stability guarantees or empirical bounds?
>
>
> We are grateful to the Reviewer for noting the absence of a formal convergence or stability analysis for the GRPO + binary reward setup. We address this point from two complementary perspectives: empirical evaluation and relevant prior work.
>
> 1. **Empirical Convergence.** In the revised appendix (Figure 5 in Appendix F), we have added  **score curves** for all four training iterations. The iterative curve can also be visualized in the README.md of our [anonymous repository](https://anonymous.4open.science/r/LoopTool). These curves consistently demonstrate:
>
> - **Stable reward score increase across iterations** - In all four training iterations, the model’s average binary reward score curves show a consistent upward trend without oscillations or divergence. This stable improvement in reward aligns with the benchmark results in Figure 2, which further confirm a steady enhancement in tool-use capability.
> - **Reward trends reflect increasing data difficulty** - As shown in Figure  5 of Appendix F, the overall difficulty of the data progressively increases as the iterations proceed. The absolute reward scores are lower in later iterations due to the increased difficulty of the data; nevertheless, the curves still exhibit steadily rising growth within each iteration’s training phase.
>
> 2. **Support from Prior Works**. The GRPO + binary reward formulation is not introduced in isolation — it has been successfully adopted in ToolRL[1] and ToolN1[2] for tool-use LLM training. While these works also do not offer a formal convergence proof, their empirical outcomes indicate practical stability in this domain. Importantly, under the same benchmark (BFCL-v3), our LoopTool-8B surpasses ToolRL and ToolN1, suggesting that our closed-loop data refinement further stabilizes the learning trajectory compared to earlier applications of the same RL paradigm.
>
> [1] ToolRL: Reward is All Tool Learning Needs, NeurIPS 2025
>
> [2] Nemotron-Research-Tool-N1: Exploring Tool-Using Language Models with Reinforced Reasoning, arxiv 2025

---

> ### Author Response · Authors · 2025-11-26
> **Request for Further Feedback on Rebuttal**
>
> Dear Reviewer N4cR,
>
> I hope this message finds you well. I have submitted detailed responses to your concerns during the rebuttal phase, and I was wondering if you have any further feedback or if there are any remaining questions regarding my submission. I would be more than happy to provide additional clarifications or address any further points you may have. I completely understand the busy review schedule, and I truly appreciate your time and effort in evaluating my work.

---

### Official Review · Reviewer_MRJu · 2025-10-25

**Soundness:** 2
**Presentation:** 3
**Contribution:** 2
**Rating:** 4
**Confidence:** 3

**Summary:**

This paper introduces LoopTool, a framework designed to improve the tool-calling capabilities of LLMs by creating a dynamic, "closed-loop" process between model training and data synthesis. The authors identify the limitations of static training datasets, which fail to adapt to a model's evolving state or correct for persistent label noise. LoopTool addresses this by iteratively: (1) diagnosing model weaknesses using Greedy Capability Probing (GCP); (2) using a judge model to verify and correct errors in both model predictions and the original dataset labels (Judgement-Guided Label Verification, JGLV); and (3) generating new, challenging data specifically targeting these identified failures (Error-Driven Data Expansion, EDDE). The authors demonstrate that an 8B model trained with this framework achieves state-of-the-art results for its scale on the BFCL-v3 and ACEBench benchmarks.

**Strengths:**

1. The experimental evaluation is a clear strength. The authors provide a validation on two relevant benchmarks (BFCL-v3 and ACEBench). The ablation studies are comprehensive and effectively isolate the contributions of each component.
2. The paper is well-written and clearly structured. The proposed LoopTool framework and its three constituent modules are explained in a logical and easy-to-follow manner.

**Weaknesses:**

1. This paper has a limited contribution. The core idea of identifying model failures, synthesizing targeted "hard" data based on those failures, correcting errors, and retraining is a very direct and intuitive workflow. This process mirrors the standard approach that many practitioners would intuitively apply when attempting to improve a model's performance on a specific, well-defined task.
2. While the authors have effectively engineered and automated this process into a "framework," the conceptual contribution feels incremental. The individual components are not novel in themselves. Error Diagnosis (GCP): Analyzing model failures is a standard part of any development cycle. Targeted Synthesis (EDDE): Using identified errors to generate new, hard samples is a well-known concept in data augmentation and curriculum learning.

**Questions:**

N/A

---

> ### Author Response · Authors · 2025-11-21
> **Response to Weakness 1&2**
>
> > Weakness 1: This paper has a limited contribution. The core idea of identifying model failures, synthesizing targeted "hard" data based on those failures, correcting errors, and retraining is a very direct and intuitive workflow.....to improve a model's performance on a specific, well-defined task.
>
>
> > Weakness 2: While the authors have effectively engineered and automated this process into a "framework," the conceptual contribution feels incremental.....data augmentation and curriculum learning.
>
> We consolidate the reviewer’s concerns into a focused response. Our reply highlights the novelty and uniqueness of the LoopTool framework from **four key perspectives**:
>
> 1. **Addressing domain‑specific challenges requires more than intuitive workflows.** In tool‑augmented LLMs, outputs must be executable API calls adhering to strict schema constraints, often in multi‑turn contexts where prior tool outputs directly influence subsequent dialogue state. **This differs sharply from error diagnosis or augmentation in tasks with purely text outputs**: Many APIs are brittle—small parameter mismatch causes execution failure; Multi‑turn dependencies mean that “errors” can propagate and compound; GRPO training in this setting suffers from reward sparsity, which means extremely easy or hard cases give zero advantage. While prior methods such as APIGen[1], ToolLLM[2], ToolACE[3], and APIGen-MT[4] have advanced synthetic data generation for tool-use, **they adopt a static, one-shot paradigm: generating large amounts of data once, followed by a single round of model training.** In such pipelines, data generation is not aware of the evolving state of the model, and cannot adapt to emerging weaknesses or overfitting tendencies.
>
> 2. **The iterative cycle is tightly coupled with the GRPO algorithm.** The GCP diagnoses the current GRPO-trained model’s mastered, borderline, and failed capabilities, using deterministic decoding and perplexity analysis. JGLV leverages evolving model predictions to systematically identify and replace erroneous annotations before they are fed into the next turn GRPO. EDDE generates structurally similar yet contextually diverse tool calls based on verified error cases, ensuring a steady stream of challenging but solvable samples. **This mutual dependency between LoopTool’s data evolution and GRPO’s learning dynamics is unique to our framework**:  The evolving dataset is aware of the RL model’s current competence, selecting and synthesizing samples that maximize GRPO’s efficiency; GRPO’s updated policy feeds back into the loop by producing outputs for judgment, enabling progressive label purification and targeted augmentation in subsequent iterations.
>
> 3. **The Experimental result is significant.** The full RL‑coupled loop yields clear, statistically and practically significant gains. The LoopTool-8B significantly surpasses its 32B data generator and achieves new state-of-the-art results on the BFCL-v3 and ACEBench benchmarks for its scale. Ablations (Table 4 in paper) show that removing coupling components consistently reduces accuracy by 2–3 points, confirming the improvements stem from the integration and feedback loop, not from the individual steps alone.
>
> 4. **LoopTool is significantly different from previous iterative frameworks.** REVERSEGEN[5] is a general-purpose failure-inducing data synthesis framework applied to diverse tasks, but it keeps the target model static during data generation; in contrast, LoopTool runs a live, GRPO-integrated closed loop. SEAL[6] implements per-instance self-edit without maintaining or refining a corpus, whereas LoopTool persistently evolves a large-scale dataset across multiple iterations. SCA[7] generates “challenging tasks” for tool-use exploration but does not produce or iteratively refine supervised datasets; LoopTool unifies difficulty-aware selection, comparative label purification, and error-driven expansion into a **dataset-centric** pipeline that yields measurable benchmark gains.
>
> In summary, LoopTool’s novelty lies in turning iterative model improvement into a closed-loop, dataset-centric process: unlike previous frameworks that either keep the model static, operate at the instance level, or focus only on task generation, LoopTool jointly evolves the model and a large-scale, purified dataset through a principled, GRPO-driven iterative pipeline.
>
> [1] APIGen: Automated Pipeline for Generating Verifiable and Diverse Function-Calling Datasets, NeurIPS 2024
>
> [2] ToolLLM: Facilitating Large Language Models to Master 16000+ Real-world APIs, ICLR 2024
>
> [3] ToolACE: Winning the Points of LLM Function Calling, ICLR 2025
>
> [4] APIGen-MT: Agentic Pipeline for Multi-Turn Data Generation via Simulated Agent-Human Interplay, NeurIPS 2025
>
> [5] Forewarned is Forearmed: Leveraging LLMs for Data Synthesis through Failure-Inducing Exploration, ICLR 2025
>
> [6] Self-Adapting Language Models, arxiv 2025
>
> [7] Self-Challenging Language Model Agents, NeurIPS 2025

---

> ### Author Response · Authors · 2025-11-26
> **Request for Further Feedback on Rebuttal**
>
> Dear Reviewer MRJu,
>
> I hope this message finds you well. I have submitted detailed responses to your concerns during the rebuttal phase, and I was wondering if you have any further feedback or if there are any remaining questions regarding my submission. I would be more than happy to provide additional clarifications or address any further points you may have. I completely understand the busy review schedule, and I truly appreciate your time and effort in evaluating my work.

---

### Official Review · Reviewer_Lr4c · 2025-11-01

**Soundness:** 3
**Presentation:** 3
**Contribution:** 2
**Rating:** 4
**Confidence:** 4

**Summary:**

LoopTool proposes a closed-loop, model-aware framework for improving tool-use capabilities of large language models (LLMs) by tightly integrating data synthesis and model training. The pipeline iteratively diagnoses model weaknesses via Greedy Capability Probing, refines noisy labels using Judgement-Guided Label Verification with an open-source judge model (Qwen3-32B), and expands hard examples through Error-Driven Data Expansion. Experiments show that an 8B model trained with LoopTool outperforms its 32B data generator and achieves state-of-the-art results on BFCL-v3 and ACEBench among models of similar scale—all without relying on closed-source APIs.

**Strengths:**

1.LoopTool introduces a fully automated, self-contained iterative pipeline that dynamically adapts training data to the model’s evolving capabilities, significantly improving tool-calling performance while avoiding costly closed-source models.
2.The framework uniquely combines label verification and error-driven data expansion in a synergistic loop, enabling both purification of noisy synthetic data and targeted generation of challenging samples, which leads to measurable gains over strong baselines.

**Weaknesses:**

1.Using Qwen3-32B as the evaluator may introduce errors due to its limited capability, potentially causing error accumulation across iterations; the paper does not adequately address how such annotation errors are mitigated over successive loops.
2.The core idea of updating training data based on model performance during iterative training has been explored in prior work such as REVERSEGEN[1]; the paper lacks a clear comparison highlighting its conceptual or technical distinctions.
3.The experiments are limited to only four iterations, with diminishing returns observed; the paper does not investigate whether a performance saturation point exists or whether further iterations could harm generalization by overfitting to hard examples.
4.The effectiveness of data generated and verified by Qwen3-32B is only evaluated on Qwen-based models; it remains unclear whether this data benefits other model families such as Llama.

[1]Forewarned is Forearmed: Leveraging LLMs for Data Synthesis through Failure-Inducing Exploration, ICLR 2025

**Questions:**

Please see weaknesses

---

> ### Author Response · Authors · 2025-11-21
> **Response to Question 1**
>
> > W1: Using Qwen3-32B as the evaluator may introduce errors due to its limited capability, potentially causing error accumulation across iterations; the paper does not adequately address how such annotation errors are mitigated over successive loops.
>
> A1: We sincerely thank you for your thorough evaluation and valuable feedback. In response to the question you raised, we provide clarifications from **three perspectives**:
>
> 1. **In JGLV, the performance of Qwen3-32B as the evaluator is highly consistent with GPT-4o.** We conducted an additional comparison in which Qwen3‑32B and GPT‑4o were separately used as evaluators in JGLV on all samples in Iterations 2–4, with the result shown in the table below. The evaluation output distributions from the two models were highly consistent (< 0.5%) for every judgment category. Furthermore, the consistency rate between the judgments of the two models exceeds 97%. The result demonstrates that Qwen3‑32B has sufficient capability and stability to serve as the judge for JGLV without introducing significant bias or drift in the iterative process. This is likely because the evaluation task, which follows a fixed and explicit instruction format, is relatively simple compared to open-ended generation. Under such well-specified criteria, Qwen3‑32B can reliably follow the instructions and thus perform consistently with GPT-4o.
>
> |             |           | Both Correct | Both Wrong | Pred Wrong | Label Wrong | Consistency Rate |
> | :---------: | :-------: | ------------ | ---------- | ---------- | ----------- | ---------------- |
> | Iteration-2 | Qwen3-32B | 14899        | 1486       | 1152       | 767         | 97.3%            |
> |             |  GPT-4o   | 14884        | 1474       | 1158       | 788         |                  |
> | Iteration-3 | Qwen3-32B | 13917        | 1001       | 2257       | 1129        | 98.8%            |
> |             |  GPT-4o   | 13882        | 1037       | 2231       | 1154        |                  |
> | Iteration-4 | Qwen3-32B | 13649        | 924        | 2734       | 997         | 97.6%            |
> |             |  GPt-4o   | 13617        | 919        | 2758       | 1010        |                  |
>
> 2. **Annotation errors are mitigated rather than accumulated through LoopTool.** Every sample entering the seed dataset has already passed a two‑tier initial verification (rule‑based API schema checks and LLM‑based contextual evaluation). Further, the risk of error accumulation is mitigated by the design of JGLV. JGLV replaces incorrect labels with higher‑quality alternatives (judged as **LABEL_WRONG**) and discards unreliable cases (judged as **BOTH_WRONG**), ensuring that the dataset becomes cleaner with each loop.
> 3. To explicitly monitor potential error accumulation, we manually sampled 200 function‑call instances from each iteration (Iterations 1–4) and performed an independent human assessment of label correctness. The correctness result is shown in the table below. This upward trend shows that annotation quality improves over successive loops, confirming that our mechanisms remove rather than amplify noise.
>
> |          | Iteration-1 | Iteration-2 | Iteration-3 | Iteration-4 |
> | -------- | ----------- | ----------- | ----------- | ----------- |
> | Accuracy | 88.5%       | 91.0%       | 93.5%       | 95.0%       |

---

> ### Author Response · Authors · 2025-11-21
> **Response to Question 2**
>
> > Q2: The core idea of updating training data based on model performance during iterative training has been explored in prior work such as REVERSEGEN[1]; the paper lacks a clear comparison highlighting its conceptual or technical distinctions.
>
> We thank the reviewer for pointing out the high‑level similarity to REVERSEGEN [1]: both methods adapt training data based on observed model weaknesses. We summarize the fundamental differences between LoopTool and REVERSEGEN from three perspectives and highlight the innovations of LoopTool. We also emphasize the distinctions between LoopTool and other contemporary works (e.g., SEAL[2], SCA[3]) in Point 4.
>
> 1. **The coupling between data evolution and model training is fundamentally different.** In REVERSEGEN, the target model itself remains static throughout the data generation process. LoopTool operates as a closed, synchronous loop, where after each GRPO‑based post‑training round, the evolving RL policy is introduced into the pipeline of data evolution. The refined dataset is immediately fed back into the next GRPO iteration, so curation is continuously aligned to the live policy’s training dynamics.
>
> 2. **The label refinement of LoopTool is distinct.** REVERSEGEN assumes the generated instructions are correct and fine‑tunes directly on them, thereby overlooking the potential noise and inaccuracies in these automatically produced labels. LoopTool explicitly purifies supervision: for each mismatch, a high‑capacity open‑source judge compares the ground‑truth label and the model’s prediction, replacing the reference with the superior output where appropriate.
>
> 3. **The application domains are entirely different.** REVERSEGEN is a general‑purpose failure‑inducing data synthesis framework applied to heterogeneous tasks such as safety red‑teaming, honesty calibration, and math reasoning. LoopTool is task‑specialized for tool‑augmented LLMs in executable, schema‑bound function‑calling benchmarks. Errors can arise from subtle schema mismatches, incorrect parameter binding, or execution‑state misalignment—even when natural‑language reasoning is correct—which makes high‑precision data synthesis and automated label verification substantially more difficult. Our closed‑loop design directly addresses these specialized constraints, which are not the focus of REVERSEGEN.
>
> 4. **LoopTool differs significantly from contemporary works such as SEAL [2] and SCA [3].** SEAL performs per‑instance self‑editing without iteratively refining a persistent dataset, so improvements remain local. SCA generates “challenging tasks” for tool‑use exploration but does not produce verifiable, schema‑compliant function‑calling data or run an RL‑aware closed loop. In contrast, LoopTool persistently evolves a large‑scale corpus across GRPO iterations, combining capability probing, judgment‑guided label correction, and error‑driven augmentation—explicitly targeting executable, multi‑turn API calls and aligning data synthesis with the model’s evolving policy to deliver measurable benchmark gains absent in SEAL or SCA.
>
> In summary,  while both methods iteratively adjust data based on model performance, LoopTool is architected as a **domain‑specific, GRPO‑integrated, self‑refining closed loop** for schema‑bound tool‑use, rather than a general failure‑search framework. Its novelty lies in the tight integration of RL training, difficulty-aware sample selection, and comparative label correction and error sample expansion—design choices absent in REVERSEGEN.
>
>
>
> [1] Forewarned is Forearmed: Leveraging LLMs for Data Synthesis through Failure-Inducing Exploration, ICLR 2025
>
> [2] Self-Adapting Language Models, arxiv 2025
>
> [3] Self-Challenging Language Model Agents, NeurIPS 2025

---

> ### Author Response · Authors · 2025-11-21
> **Response to Question 3**
>
> > Q3: The experiments are limited to only four iterations, with diminishing returns observed; the paper does not investigate whether a performance saturation point exists or whether further iterations could harm generalization by overfitting to hard examples.
>
> In response to the question you raised, we conduct additional iterations beyond those reported to investigate the saturation behavior and potential overfitting risks. Specifically, for **LoopTool‑8B**, we extended training by two further iterations under identical settings. The results remained relatively stable, changing from 74.93% (Iteration 4) to 74.62% (Iteration 5), and then to 74.24% (Iteration 6). The same trend is corroborated in our LLaMA‑based experiments (see response to Question 4), where performance peaks at Iteration 3 (61.00%), then saturates and slightly declines after Iteration 4 (60.32%).
>
> | LoopTool-8B | Overall | Non-Live Acc | Live Acc | Multi-Turn Acc |
> | :---------: | :-----: | :----------: | :------: | :------------: |
> | Iteration-4 |  74.93  |    89.52     |  84.72   |     50.88      |
> | Iteration-5 |  74.62  |    88.12     |  83.79   |     50.12      |
> | Iteration-6 |  74.24  |    87.96     |  83.65   |     50.34      |
>
> We attribute this decline to model capacity limits — the 8B backbone may have reached its representational ceiling, such that continued injection of increasingly hard samples does not yield further gains. These empirical observations clarify that **a performance saturation point does exist** for our current configuration, and extending iterations beyond this point induces mild overfitting.

---

> ### Author Response · Authors · 2025-11-21
> **Response to Question 4**
>
> > Q4: The effectiveness of data generated and verified by Qwen3-32B is only evaluated on Qwen-based models; it remains unclear whether this data benefits other model families, such as Llama.
>
> We thank the reviewer for raising the concern regarding the generality of data generated and verified by Qwen3‑32B beyond Qwen‑based models. LoopTool is essentially a **model-agnostic** framework for iterative evolution of data and models. To directly address this concern, we conducted additional experiments using **Llama‑3.1‑8B‑Instruct** as the learned policy, while keeping training budget, number of steps, and all LoopTool configurations identical to the Qwen‑based runs. We still employ the Qwen3-32B model as both the Generator and the Evaluator. The results are illustrated in the table below. Results show clear **iterative improvements**: overall accuracy rises from 49.72% (Original) to 61.00% (Iteration 3), with substantial metric‑level gains such as **+8.51 points in Multi‑Turn accuracy** and **+35.37 points in Irrelevance**. Performance plateaus by Iteration 4 (60.82%), suggesting convergence.
>
> | Llama-3.1-8B-Instruct | Overall | Non-Live Acc | Live Acc | Multi-Turn Acc | Irrelevance |
> | :-------------------: | :-----: | :----------: | :------: | :------------: | :---------: |
> |       Original        |  49.72  |    84.48     |  61.13   |      9.62      |    48.46    |
> |      Iteration-1      |  53.99  |    85.46     |  71.97   |     11.75      |    75.29    |
> |      Iteration-2      |  56.39  |    86.17     |  74.14   |     14.12      |    82.55    |
> |      Iteration-3      |  61.00  |    86.73     |  77.74   |     18.13      |    83.83    |
> |      Iteration-4      |  60.32  |    87.04     |  75.66   |     17.50      |    85.78    |
>
> These results provide direct evidence that the LoopTool framework offers substantial benefits to a different model family (Llama), demonstrating that the improvements are not specific to Qwen‑based models. In this newly updated revision, we supplement the iterative experiments of the Llama-3.1-8B model in Appendix E.

---

> ### Author Response · Authors · 2025-11-26
> **Request for Further Feedback on Rebuttal**
>
> Dear Reviewer Lr4c,
>
> I hope this message finds you well. I have submitted detailed responses to your concerns during the rebuttal phase, and I was wondering if you have any further feedback or if there are any remaining questions regarding my submission. I would be more than happy to provide additional clarifications or address any further points you may have. I completely understand the busy review schedule, and I truly appreciate your time and effort in evaluating my work.

---

### Note · Authors · 2025-12-26

I have read and agree with the venue's withdrawal policy on behalf of myself and my co-authors.